# Replicability in Reinforcement Learning[*]

**Amin Karbasi**
Yale University, Google Research
`amin.karbasi@yale.edu`

**Grigoris Velegkas**
Yale University
`grigoris.velegkas@yale.edu`

**Lin F. Yang**
UCLA
`linyang@ee.ucla.edu`

**Felix Zhou**
Yale University
`felix.zhou@yale.edu`

## Abstract

We initiate the mathematical study of replicability as an algorithmic property in the context of reinforcement learning (RL). We focus on the fundamental setting of discounted tabular MDPs with access to a *generative model*. Inspired by Impagliazzo et al. [2022], we say that an RL algorithm is replicable if, with high probability, it outputs the *exact* same policy after two executions on i.i.d. samples drawn from the generator when its *internal* randomness is the same. We first provide an efficient $\rho$-replicable algorithm for $(\varepsilon, \delta)$-optimal policy estimation with sample and time complexity $\widetilde{O}\left(\frac{N^3 \cdot \log(1/\delta)}{(1-\gamma)^5 \cdot \varepsilon^2 \cdot \rho^2}\right)$, where $N$ is the number of state-action pairs. Next, for the subclass of deterministic algorithms, we provide a lower bound of order $\Omega\left(\frac{N^3}{(1-\gamma)^3 \cdot \varepsilon^2 \cdot \rho^2}\right)$. Then, we study a relaxed version of replicability proposed by Kalavasis et al. [2023] called TV *indistinguishability*. We design a computationally efficient TV indistinguishable algorithm for policy estimation whose sample complexity is $\widetilde{O}\left(\frac{N^2 \cdot \log(1/\delta)}{(1-\gamma)^5 \cdot \varepsilon^2 \cdot \rho^2}\right)$. At the cost of $\exp(N)$ running time, we transform these TV indistinguishable algorithms to $\rho$-replicable ones without increasing their sample complexity. Finally, we introduce the notion of *approximate*-replicability where we only require that two outputted policies are close under an appropriate statistical divergence (e.g., Renyi) and show an improved sample complexity of $\widetilde{O}\left(\frac{N \cdot \log(1/\delta)}{(1-\gamma)^5 \cdot \varepsilon^2 \cdot \rho^2}\right)$.

## 1 Introduction

When designing a reinforcement learning (RL) algorithm, how can one ensure that when it is executed twice in the same environment its outcome will be the same? In this work, our goal is to design RL algorithms with *provable* replicability guarantees. The lack of replicability in scientific research, which the community also refers to as the *reproducibility crisis*, has been a major recent concern. This can be witnessed by an article that appeared in Nature [Baker, 2016]: Among the 1,500 scientists who participated in a survey, 70% of them could not replicate other researchers' findings and more shockingly, 50% of them could not even reproduce their own results. Unfortunately, due to the exponential increase in the volume of Machine Learning (ML) papers that are being published each year, the ML community has also observed an alarming increase in the lack of reproducibility. As a result, major ML conferences such as NeurIPS and ICLR have established "reproducibility challenges" in which researchers are encouraged to replicate the findings of their colleagues [Pineau et al., 2019, 2021].

---

[*]Authors are listed alphabetically.

37th Conference on Neural Information Processing Systems (NeurIPS 2023).

Recently, RL algorithms have been a crucial component of many ML systems that are being deployed in various application domains. These include but are not limited to, competing with humans in games [Mnih et al., 2013, Silver et al., 2017, Vinyals et al., 2019, , FAIR], creating self-driving cars [Kiran et al., 2021], designing recommendation systems [Afsar et al., 2022], providing e-healthcare services [Yu et al., 2021], and training Large Language Models (LLMs) [Ouyang et al., 2022]. In order to ensure replicability across these systems, an important first step is to develop replicable RL algorithms. To the best of our knowledge, replicability in the context of RL has not received a formal mathematical treatment. We initiate this effort by focusing on *infinite horizon, tabular* RL with a *generative model*. The generative model was first studied by Kearns and Singh [1998] in order to understand the statistical complexity of long-term planning without the complication of exploration. The crucial difference between this setting and Dynamic Programming (DP) [Bertsekas, 1976] is that the agent needs to first obtain information about the world before computing a *policy* through some optimization process. Thus, the main question is to understand the number of samples required to estimate a near-optimal policy. This problem is similar to understanding the number of labeled examples required in PAC learning [Valiant, 1984].

In this work, we study three different formal notions of replicability and design algorithms that satisfy them. First, we study the definition of Impagliazzo et al. [2022], which adapted to the context of RL says that a learning algorithm is replicable if it outputs the exact same policy when executed twice on the same MDP, using *shared* internal randomness across the two executions (cf. Definition 2.10). We show that there exists a replicable algorithm that outputs a near-optimal policy using $\widetilde{O}(N^3)$ samples[2], where $N$ is the cardinality of the state-action space. This algorithm satisfies an additional property we call *locally random*, which roughly asks that every random decision the algorithm makes based on internal randomness must draw its internal randomness independently from other decisions. Next, we provide a lower bound for deterministic algorithms that matches this upper bound.

Subsequently, we study a less stringent notion of replicability called TV indistinguishability, which was introduced by Kalavasis et al. [2023]. This definition states that, in expectation over the random draws of the input, the TV distance of the two distributions over the outputs of the algorithm should be small (cf. Definition 4.1). We design a computationally efficient TV indistinguishable algorithm for answering $d$ statistical queries whose sample complexity scales as $\widetilde{O}(d^2)$. We remark that this improves the sample complexity of its replicable counterpart based on the rounding trick from Impagliazzo et al. [2022] by a factor of $d$ and it has applications outside the scope of our work [Impagliazzo et al., 2022, Esfandiari et al., 2023b,a, Bun et al., 2023, Kalavasis et al., 2023]. This algorithm is inspired by the Gaussian mechanism from the Differential Privacy (DP) literature [Dwork et al., 2014]. Building upon this statistical query estimation oracle, we design computationally efficient TV-indistinguishable algorithms for $Q$-function estimation and policy estimation whose sample complexity scales as $\widetilde{O}(N^2)$. Interestingly, we show that by violating the locally random property and allowing for internal randomness that creates correlations across decisions, we can transform these TV indistinguishable algorithms to replicable ones without hurting their sample complexity, albeit at a cost of $\widetilde{O}(\exp(N))$ running time. Our transformation is inspired by the main result of Kalavasis et al. [2023]. We also conjecture that the true sample complexity of $\rho$-replicable policy estimation is indeed $\widetilde{\Theta}(N^2)$.

Finally, we propose a novel relaxation of the previous notions of replicability. Roughly speaking, we say that an algorithm is *approximately replicable* if, with high probability, when executed twice on the same MDP, it outputs policies that are close under a dissimilarity measure that is based on the *Renyi divergence*. We remark that this definition does not require sharing the internal randomness across the executions. Finally, we design an RL algorithm that is approximately replicable and outputs a near-optimal policy with $\widetilde{O}(N)$ sample and time complexity.

Table 1.1 and Table 1.2 summarizes the sample and time complexity of $Q$-estimation and policy estimation, respectively, under different notions of replicability. We assume the algorithms in question have a constant probability of success. In Appendix G, we further discuss the benefits and downsides for each of these notions.

---

[2]For simplicity, we hide the dependence on the remaining parameters of the problem in this section.

Table 1.1: Complexity Overview for $Q$-Estimation with Constant Probability of Success.

| PROPERTY | SAMPLE COMPLEXITY | TIME COMPLEXITY |
|---|---|---|
| LOCALLY RANDOM, REPLICABLE | $\tilde{\Theta}\left(\frac{N^3}{(1-\gamma)^3\varepsilon^2\rho^2}\right)$ | $\tilde{\Theta}\left(\frac{N^3}{(1-\gamma)^3\varepsilon^2\rho^2}\right)$ |
| TV INDISTINGUISHABLE | $\tilde{O}\left(\frac{N^2}{(1-\gamma)^3\varepsilon^2\rho^2}\right)$ | $\tilde{O}\left(\frac{\mathrm{poly}(N)}{(1-\gamma)^3\varepsilon^2\rho^2}\right)$ |
| REPLICABLE (THROUGH TV INDISTINGUISHABILITY) | $\tilde{O}\left(\frac{N^2}{(1-\gamma)^3\varepsilon^2\rho^2}\right)$ | $\tilde{O}\left(\frac{\exp(N)}{(1-\gamma)^3\varepsilon^2\rho^2}\right)$ |

Table 1.2: Complexity Overview for Policy Estimation with Constant Probability of Success.

| PROPERTY | SAMPLE COMPLEXITY | TIME COMPLEXITY |
|---|---|---|
| LOCALLY RANDOM, REPLICABLE | $\tilde{O}\left(\frac{N^3}{(1-\gamma)^5\varepsilon^2\rho^2}\right)$ | $\tilde{O}\left(\frac{N^3}{(1-\gamma)^5\varepsilon^2\rho^2}\right)$ |
| TV INDISTINGUISHABLE | $\tilde{O}\left(\frac{N^2}{(1-\gamma)^5\varepsilon^2\rho^2}\right)$ | $\tilde{O}\left(\frac{\mathrm{poly}(N)}{(1-\gamma)^5\varepsilon^2\rho^2}\right)$ |
| REPLICABLE (THROUGH TV INDISTINGUISHABILITY) | $\tilde{O}\left(\frac{N^2}{(1-\gamma)^5\varepsilon^2\rho^2}\right)$ | $\tilde{O}\left(\frac{\exp(N)}{(1-\gamma)^5\varepsilon^2\rho^2}\right)$ |
| APPROXIMATELY REPLICABLE | $\tilde{O}\left(\frac{N}{(1-\gamma)^5\varepsilon^2\rho^2}\right)$ | $\tilde{O}\left(\frac{N}{(1-\gamma)^5\varepsilon^2\rho^2}\right)$ |

## 1.1 Related Works

**Replicability.** Pioneered by Impagliazzo et al. [2022], there has been a growing interest from the learning theory community in studying replicability as an algorithmic property. Esfandiari et al. [2023a,b] studied replicable algorithms in the context of multi-armed bandits and clustering. Recently, Bun et al. [2023] established equivalences between replicability and other notions of algorithmic stability such as differential privacy when the domain of the learning problem is finite and provided some computational and statistical hardness results to obtain these equivalences, under cryptographic assumptions. Subsequently, Kalavasis et al. [2023] proposed a relaxation of the replicability definition of Impagliazzo et al. [2022], showed its statistical equivalence to the notion of replicability for countable domains[3] and extended some of the equivalences from Bun et al. [2023] to countable domains. Chase et al. [2023], Dixon et al. [2023] proposed a notion of *list-replicability*, where the output of the learner is not necessarily identical across two executions but is limited to a small list of choices.

The closest related work to ours is the concurrent and independent work of Eaton et al. [2023]. They also study a formal notion of replicability in RL which is inspired by the work of Impagliazzo et al. [2022] and coincides with one of the replicability definitions we are studying (cf. Definition 2.8). Their work focuses both on the generative model and the episodic exploration settings. They derive upper bounds on the sample complexity in both settings and validate their results experimentally. On the other hand, our work focuses solely on the setting with the generative model. We obtain similar sample complexity upper bounds for replicable RL algorithms under Definition 2.8 and then we show a lower bound for the class of locally random algorithms. Subsequently, we consider two relaxed notions of replicability which yield improved sample complexities.

**Reproducibility in RL.** Reproducing, interpreting, and evaluating empirical results in RL can be challenging since there are many sources of randomness in standard benchmark environments. Khetarpal et al. [2018] proposed a framework for evaluating RL to improve reproducibility. Another barrier to reproducibility is the unavailability of code and training details within technical reports. Indeed, Henderson et al. [2018] observed that both intrinsic (e.g. random seeds, environments) and extrinsic (e.g. hyperparameters, codebases) factors can contribute to difficulties in reproducibility. Tian et al. [2019] provided an open-source implementation of AlphaZero [Silver et al., 2017], a popular RL-based Go engine.

---

[3]We remark that this equivalence for finite domains can also be obtained, implicitly, from the results of Bun et al. [2023].

**RL with a Generative Model.** The study of RL with a generative model was initiated by Kearns and Singh [1998] who provided algorithms with suboptimal sample complexity in the discount factor $\gamma$. A long line of work (see, e.g. Gheshlaghi Azar et al. [2013], Wang [2017], Sidford et al. [2018a,b], Feng et al. [2019], Agarwal et al. [2020], Li et al. [2020] and references therein) has led to (non-replicable) algorithms with minimax optimal sample complexity. Another relevant line of work that culminated with the results of Even-Dar et al. [2002], Mannor and Tsitsiklis [2004] studied the sample complexity of finding an $\varepsilon$-optimal arm in the multi-armed bandit setting with access to a generative model.

## 2 Setting

### 2.1 Reinforcement Learning Setting

**(Discounted) Markov Decision Process.** We start by providing the definitions related to the *Markov Decision Process* (MDP) that we study in this work.

**Definition 2.1** (Discounted Markov Decision Process). A *(discounted) Markov decision process (MDP)* is a 6-tuple $M = \left(\mathcal{S}, s_0, \mathcal{A} = \bigcup_{s \in \mathcal{S}} \mathcal{A}^s, P_M, r_M, \gamma\right)$. Here $\mathcal{S}$ is a finite set of states, $s_0 \in \mathcal{S}$ is the initial state, $\mathcal{A}^s$ is the finite set of available actions for state $s \in \mathcal{S}$, and $P_M(s' \mid s, a)$ is the transition kernel, i.e, $\forall (s, s') \in \mathcal{S}^2, \forall a \in \mathcal{A}^s, P_M(s' \mid s, a) \geq 0$ and $\forall s \in \mathcal{S}, \forall a \in \mathcal{A}^s, \sum_{s' \in \mathcal{S}} P_M(s' \mid s, a) = 1$. We denote the reward function[4] by $r_M : \mathcal{S} \times \mathcal{A} \rightarrow [0, 1]$ and the discount factor by $\gamma \in (0, 1)$. The interaction between the agent and the environment works as follows. At every step, the agent observes a state $s$ and selects an action $a \in \mathcal{A}^s$, yielding an instant reward $r_M(s, a)$. The environment then transitions to a random new state $s' \in \mathcal{S}$ drawn according to the distribution $P_M(\cdot \mid s, a)$.

**Definition 2.2** (Policy). We say that a map $\pi : \mathcal{S} \rightarrow \mathcal{A}$ is a *(deterministic) stationary policy*.

When we consider randomized policies we overload the notation and denote $\pi(s, a)$ the probability mass that policy $\pi$ puts on action $a \in \mathcal{A}^s$ in state $s \in \mathcal{S}$.

**Definition 2.3** (Value $(V)$ Function). The *value $(V)$ function* $V_M^\pi : \mathcal{S} \rightarrow [0, 1/(1-\gamma)]$ of a policy $\pi$ with respect to the MDP $M$ is given by $V_M^\pi(s) := \mathbb{E}\left[\sum_{t=0}^\infty \gamma^t r_M(s_t, a_t) \mid s_0 = s\right]$. Here $a_t \sim \pi(s_t)$ and $s_{t+1} \sim P_M(\cdot \mid s_t, a_t)$.

This is the expected discounted cumulative reward of a policy.

**Definition 2.4** (Action-Value $(Q)$ Function). The *action-value $(Q)$ function* $Q_M^\pi : \mathcal{S} \times \mathcal{A} \rightarrow [0, 1/(1-\gamma)]$ of a policy $\pi$ with respect to the MDP $M$ is given by $Q_M^\pi(s, a) := r_M(s, a) + \gamma \cdot \sum_{s' \in \mathcal{S}} P_M(s' \mid s, a) \cdot V_M^\pi(s')$.

We write $N := \sum_{s \in \mathcal{S}} |\mathcal{A}^s|$ to denote the number of state-action pairs. We denote by $\pi^\star$ the *optimal* policy that maximizes the value function, i.e., $\forall \pi, s \in \mathcal{S}: V^\star(s) := V^{\pi^\star}(s) \geq V^\pi(s)$. We also define $Q^\star(s, a) := Q^{\pi^\star}(s, a)$. This quantity is well defined since the fundamental theorem of RL states that there exists a (deterministic) policy $\pi^\star$ that simultaneously maximizes $V^\pi(s)$ among all policies $\pi$, for all $s \in \mathcal{S}$ (see e.g. Puterman [2014]).

Since estimating the optimal policy from samples when $M$ is unknown could be an impossible task, we aim to compute an $\varepsilon$-*approximately* optimal policy for $M$.

**Definition 2.5** (Approximately Optimal Policy). Let $\varepsilon \in (0, 1)$. We say that the policy $\pi$ is $\varepsilon$-approximately optimal if $||V^\star - V^\pi||_\infty \leq \varepsilon$.

In the above definition, $|| \cdot ||_\infty$ denotes the infinity norm of the vector, i.e., its maximum element in absolute value.

**Generative Model.** Throughout this work, we assume we have access to a *generative model* (first studied in Kearns and Singh [1998]) or a *sampler* $G_M$, which takes as input a state-action pair $(s, a)$ and provides a sample $s' \sim P_M(\cdot \mid s, a)$. This widely studied fundamental RL setting allows us to focus on the sample complexity of planning over a long horizon without considering the additional complications of exploration. Since our focus throughout this paper is on the *statistical* complexity of

---

[4]We assume that the reward is deterministic and known to the learner. Our results hold for stochastic and unknown rewards with an extra (replicable) estimation step, which does not increase the overall sample complexity.

the problem, our goal is to achieve the desired algorithmic performance while minimizing the number of samples from the generator that the algorithm requires.

**Approximately Optimal Policy Estimator.** We now define what it means for an algorithm $\mathscr{A}$ to be an approximately optimal policy estimator.

**Definition 2.6** $((\varepsilon, \delta)$-Optimal Policy Estimator). Let $\varepsilon, \delta \in (0,1)^2$. A (randomized) algorithm $\mathscr{A}$ is called an $(\varepsilon, \delta)$-optimal policy estimator if there exists a number $n := n(\varepsilon, \delta) \in \mathbb{N}$ such that, for any MDP $M$, when it is given at least $n(\varepsilon, \delta)$ samples from the generator $G_M$, it outputs a policy $\hat{\pi}$ such that $\left\| V^{\hat{\pi}} - V^\star \right\|_\infty \leq \varepsilon$ with probability at least $1 - \delta$. Here, the probability is over random draws from $G_M$ and the internal randomness of $\mathscr{A}$.

Approximately optimal $V$-function estimators and $Q$-function estimators are defined similarly.

*Remark* 2.7. In order to allow flexibility to the algorithm, we do not restrict it to request the same amount of samples for every state-action pair. Thus $n(\varepsilon, \delta)$ is a bound on the total number of samples that $\mathscr{A}$ receives from $G_M$. The algorithms we design request the same number of samples for every state-action pair, however, our lower bounds are stronger and hold without this restriction.

When the MDP $M$ is clear from context, we omit the subscript in all the previous quantities.

## 2.2 Replicability

**Definition 2.8** (Replicable Algorithm; [Impagliazzo et al., 2022]). Let $\mathscr{A} : \mathcal{I}^n \to \mathcal{O}$ be an $n$-sample randomized algorithm that takes as input elements from some domain $\mathcal{I}$ and maps them to some co-domain $\mathcal{O}$. Let $\mathcal{R}$ denote the internal distribution over binary strings that $\mathscr{A}$ uses. For $\rho \in (0,1)$, we say that $\mathscr{A}$ is $\rho$-*replicable* if for any distribution $\mathcal{D}$ over $\mathcal{I}$ it holds that $\mathbb{P}_{\bar{S}, \bar{S}' \sim \mathcal{D}^n, \bar{r} \sim \mathcal{R}} \left\{ \mathscr{A}(\bar{S}; \bar{r}) = \mathscr{A}(\bar{S}'; \bar{r}) \right\} \geq 1 - \rho$, where $\mathscr{A}(\bar{S}; \bar{r})$ denotes the (deterministic) output of $\mathscr{A}$ when its input is $\bar{S}$ and the realization of the internal random string is $\bar{r}$.

In the context of our work, we should think of $\mathscr{A}$ as a randomized mapping that receives samples from the generator $G$ and outputs policies. Thus, even when $\bar{S}$ is fixed, $\mathscr{A}(\bar{S})$ should be thought of as a random variable, whereas $\mathscr{A}(\bar{S}; \bar{r})$ is the *realization* of this variable given the (fixed) $\bar{S}, \bar{r}$. We should think of $\bar{r}$ as the shared randomness between the two executions, which can be implemented as a shared random seed.

One of the most elementary statistical operations we may wish to make replicable is mean estimation. This operation can be phrased using the language of *statistical queries*.

**Definition 2.9** (Statistical Query Oracle; [Kearns, 1998]). Let $\mathcal{D}$ be a distribution over the domain $\mathcal{X}$ and $\phi : \mathcal{X}^n \to \mathbb{R}$ be a statistical query with true value $v^\star := \lim_{n \to \infty} \phi(X_1, \ldots, X_n) \in \mathbb{R}$. Here $X_i \sim_{i.i.d.} \mathcal{D}$ and the convergence is understood in probability or distribution. Let $\varepsilon, \delta \in (0,1)^2$. A *statistical query (SQ) oracle* outputs a value $v$ such that $|v - v^\star| \leq \varepsilon$ with probability at least $1 - \delta$.

The simplest example of a statistical query is the sample mean $\phi(X_1, \ldots, X_n) = \frac{1}{n} \sum_{i=1}^n X_i$. Impagliazzo et al. [2022] designed a replicable SQ-query oracle for sample mean queries with bounded co-domain (cf. Theorem C.1).

The following definition is the formal instantiation of Definition 2.8 in the setting we are studying.

**Definition 2.10** (Replicable Policy Estimator). Let $\rho \in (0,1)$. A policy estimator $\mathscr{A}$ that receives samples from a generator $G$ and returns a policy $\pi$ using internal randomness $\mathcal{R}$ is $\rho$-replicable if for any MDP $M$, when two sequences of samples $\bar{S}, \bar{S}'$ are generated independently from $G$, it holds that $\mathbb{P}_{\bar{S}, \bar{S}' \sim G, \bar{r} \sim \mathcal{R}} \left\{ \mathscr{A}(\bar{S}; \bar{r}) = \mathscr{A}(\bar{S}'; \bar{r}) \right\} \geq 1 - \rho$.

To give the reader some intuition about the type of problems for which replicable algorithms under Definition 2.8 exist, we consider the fundamental task of estimating the mean of a random variable. Impagliazzo et al. [2022] provided a replicable mean estimation algorithm when the variable is bounded (cf. Theorem C.1). Esfandiari et al. [2023b] generalized the result to simultaneously estimate the means of multiple random variables with unbounded co-domain under some regularity conditions on their distributions (cf. Theorem C.2). The idea behind both results is to use a rounding trick introduced in Impagliazzo et al. [2022] which allows one to sacrifice some accuracy of the estimator in favor of the replicability property. The formal statement of both results, which are useful for our work, are deferred to Appendix C.1.

# 3 Replicable $Q$-Function & Policy Estimation

Our aim in this section is to understand the sample complexity overhead that the replicability property imposes on the task of computing an $(\varepsilon, \delta)$- approximately optimal policy. Without this requirement, Sidford et al. [2018a], Agarwal et al. [2020], Li et al. [2020] showed that $\widetilde{O}(N \log(1/\delta)/[(1-\gamma)^3 \varepsilon^2])$ samples suffice to estimate such a policy, value function, and $Q$-function. Moreover, since Gheshlaghi Azar et al. [2013] provided matching lower bounds[5], the sample complexity for this problem has been settled. Our main results in this section are tight sample complexity bounds for locally random $\rho$-replicable $(\varepsilon, \delta)$-approximately optimal $Q$-function estimation as well as upper and lower bounds for $\rho$-replicable $(\varepsilon, \delta)$-approximately policy estimation that differ by a factor of $1/(1-\gamma)^2$. The missing proofs for this section can be found in Appendix D.

We remark that in both the presented algorithms and lower bounds, we assume local randomness. For example, we assume that the internal randomness is drawn independently for each state-action pair for replicable $Q$-estimation. In the case where we allow for the internal randomness to be correlated across estimated quantities, we present an algorithm that overcomes our present lower bound in Section 4.3. However, the running time of this algorithm is exponential in $N$.

## 3.1 Computationally Efficient Upper Bound on the Sample Complexity

We begin by providing upper bounds on the sample complexity for replicable estimation of an approximately optimal policy and $Q$-function. On a high level, we follow a two-step approach: 1) Start with black-box access to some $Q$-estimation algorithm that is not necessarily replicable (cf. Theorem D.2) to estimate some $\widehat{Q}$ such that $\|Q^\star - \widehat{Q}\|_\infty \leq \varepsilon_0$. 2) Apply the replicable rounding algorithm from Theorem C.2 as a post-processing step. The rounding step incurs some loss of accuracy in the estimated $Q$-function. Therefore, in order to balance between $\rho$-replicability and $(\varepsilon, \delta)$-accuracy, we need to call the black-box oracle with an accuracy smaller than $\varepsilon$, i.e. choose $\varepsilon_0 < O(\varepsilon\rho)$. This yields an increase in the sample complexity which we quantify below. For the proof details, see Appendix D.1.

Recall that $N$ is the number of state-action pairs of the MDP.

**Theorem 3.1.** *Let $\varepsilon, \rho \in (0,1)^2$ and $\delta \in (0, \rho/3)$. There is a locally random $\rho$-replicable algorithm that outputs an $\varepsilon$-optimal $Q$-function with probability at least $1 - \delta$. Moreover, it has time and sample complexity $\widetilde{O}(N^3 \log(1/\delta)/[(1-\gamma)^3 \varepsilon^2 \rho^2])$.*

So far, we have provided a replicable algorithm that outputs an approximately optimal $Q$ function. The main result of Singh and Yee [1994] shows that if $\|\widehat{Q} - Q^\star\|_\infty \leq \varepsilon$, then the greedy policy with respect to $\widehat{Q}$, i.e., $\forall s \in \mathcal{S}, \widehat{\pi}(s) := \arg\max_{a \in \mathcal{A}^s} \widehat{Q}(s, a)$, is $\varepsilon/(1-\gamma)$-approximately optimal (cf. Theorem D.3). Thus, if we want to obtain an $\varepsilon$-approximately optimal policy, it suffices to obtain a $(1-\gamma)\varepsilon$-approximately optimal $Q$-function. This is formalized in Corollary 3.2.

**Corollary 3.2.** *Let $\varepsilon, \rho \in (0,1)^2$ and $\delta \in (0, \rho/3)$. There is a locally random $\rho$-replicable algorithm that outputs an $\varepsilon$-optimal policy with probability at least $1 - \delta$. Moreover, it has time and sample complexity $\widetilde{O}(N^3 \log(1/\delta)/[(1-\gamma)^5 \varepsilon^2 \rho^2])$.*

Again, we defer the proof to Appendix D.1.

Due to space limitation, the lower bound derivation is postponed to Appendix D.

# 4 TV Indistinguishable Algorithms for $Q$-Function and Policy Estimation

In this section, we present an algorithm with an improved sample complexity for replicable $Q$-function estimation and policy estimation. Our approach consists of several steps. First, we design a computationally efficient SQ algorithm for answering $d$ statistical queries that satisfies the *total variation (TV) indistinguishability* property [Kalavasis et al., 2023] (cf. Definition 4.1), which can be viewed as a relaxation of replicability. The new SQ algorithm has an improved sample complexity compared to its replicable counterpart we discussed previously. Using this oracle, we show how we can design computationally efficient $Q$-function estimation and policy estimation algorithms that

---

[5]up to logarithmic factors

satisfy the TV indistinguishability definition and have an improved sample complexity by a factor of $N$ compared to the ones in Section 3.1. Then, by describing a specific implementation of its *internal* randomness, we make the algorithm replicable. Unfortunately, this step incurs an exponential cost in the computational complexity of the algorithm with respect to the cardinality of the state-action space. We emphasize that the reason we are able to circumvent the lower bound of Appendix D.2 is that we use a specific source of internal randomness that creates correlations across the random choices of the learner. Our result reaffirms the observation made by Kalavasis et al. [2023] that the same learning algorithm, i.e., input $\rightarrow$ output mapping, can be replicable under one implementation of its internal randomness but not replicable under a different one.

First, we state the definition of TV indistinguishability from Kalavasis et al. [2023].

**Definition 4.1** (TV Indistinguishability; [Kalavasis et al., 2023])**.** A learning rule $\mathscr{A}$ is $n$-sample $\rho$-TV indistinguishable if for any distribution over inputs $\mathcal{D}$ and two independent samples $S, S' \sim \mathcal{D}^n$ it holds that $\mathbb{E}_{S,S'\sim\mathcal{D}^n}[d_{\mathrm{TV}}(A(S), A(S'))] \leq \rho$.

In their work, Kalavasis et al. [2023] showed how to transform any $\rho$-TV indistinguishable algorithm to a $2\rho/(1+\rho)$-replicable one when the input domain is *countable*. Importantly, this transformation does not change the input $\rightarrow$ output mapping that is induced by the algorithm. A similar transformation for finite domains can also be obtained by the results in Bun et al. [2023]. We emphasize that neither of these two transformations are computationally efficient. Moreover, Bun et al. [2023] give cryptographic evidence that there might be an inherent computational hardness to obtain the transformation.

## 4.1 TV Indistinguishable Estimation of Multiple Statistical Queries

We are now ready to present a TV-indistinguishable algorithm for estimating $d$ independent statistical queries. The high-level approach is as follows. First, we estimate each statistical query up to accuracy $\varepsilon\rho/\sqrt{d}$ using black-box access to the SQ oracle and we get an estimate $\widehat{\mu}_1 \in [0,1]^d$. Then, the output of the algorithm is drawn from $\mathcal{N}(\widehat{\mu}_1, \varepsilon^2 I_d)$. Since the estimated mean of each query is accurate up to $\varepsilon\rho/\sqrt{d}$ and the variance is $\varepsilon^2$, we can see that, with high probability, the estimate of each query will be accurate up to $O(\varepsilon)$. To argue about the TV indistinguishability property, we first notice that, with high probability across the two executions, the estimate $\widehat{\mu}_2 \in [0,1]^d$ satisfies $||\widehat{\mu}_1 - \widehat{\mu}_2||_\infty \leq 2\rho \cdot \varepsilon/\sqrt{d}$. Then, we can bound the TV distance of the output of the algorithm as $d_{\mathrm{TV}}\left(\mathcal{N}(\widehat{\mu}_1, \varepsilon^2 I_d), \mathcal{N}(\widehat{\mu}_2, \varepsilon^2 I_d)\right) \leq O(\rho)$ [Gupta, 2020]. We underline that this behavior is reminiscent of the advanced composition theorem in the Differential Privacy (DP) literature (see e.g., Dwork et al. [2014]) and our algorithm can be viewed as an extension of the Gaussian mechanism from the DP line of work to the replicability setting. This algorithm has applications outside the scope of our work since multiple statistical query estimation is a subroutine widely used in the replicability line of work [Impagliazzo et al., 2022, Esfandiari et al., 2023b,a, Bun et al., 2023, Kalavasis et al., 2023]. This discussion is formalized in the following theorem.

**Theorem 4.2** (TV Indistinguishable SQ Oracle for Multiple Queries)**.** *Let $\varepsilon, \rho \in (0,1)^2$ and $\delta \in (0, \rho/5)$. Let $\phi_1, \ldots, \phi_d$ be $d$ statistical queries with co-domain $[0,1]$. Assume that we can simultaneously estimate the true values of all $\phi_i$'s with accuracy $\varepsilon$ and confidence $\delta$ using $n(\varepsilon, \delta)$ total samples. Then, there exists a $\rho$-TV indistinguishable algorithm (Algorithm A.1) that requires at most $n(\varepsilon\rho/[2\sqrt{8d\cdot\log(4d/\delta)}], \delta/2)$ many samples to output estimates $\widehat{v}_1, \ldots, \widehat{v}_d$ of the true values $v_1, \ldots, v_d$ to guarantee that $\max_{i\in[d]}|\widehat{v}_i - v_i| \leq \varepsilon$, with probability at least $1 - \delta$.*

## 4.2 TV Indistinguishable $Q$-Function and Policy Estimation

Equipped with Algorithm A.1, we are now ready to present a TV-indistinguishable algorithm for $Q$-function estimation and policy estimation with superior sample complexity compared to the one in Section 3.1. The idea is similar to the one in Section 3.1. We start with black-box access to an algorithm for $Q$-function estimation, and then we apply the Gaussian mechanism (Algorithm A.1). We remark that the running time of this algorithm is polynomial in all the parameters of the problem.

Recall that $N$ is the number of state-action pairs of the MDP.

**Theorem 4.3.** *Let $\varepsilon, \rho \in (0,1)^2$ and $\delta \in (0, \rho/5)$. There is a $\rho$-TV indistinguishable algorithm that outputs an $\varepsilon$-optimal $Q$-function with probability at least $1 - \delta$. Moreover, it has time and sample complexity $\widetilde{O}(N^2 \log(1/\delta)/[(1-\gamma)^3\varepsilon^2\rho^2])$.*

*Proof.* The proof follows by combining the guarantees of Sidford et al. [2018a] (Theorem D.2) and Theorem 4.2. To be more precise, Theorem D.2 shows that in order to compute some $\widehat{Q}$ such that $\|\widehat{Q} - Q\|_\infty \le \varepsilon$, one needs $\widetilde{O}(N \log(1/\delta)/[(1-\gamma)^3\varepsilon^2])$. Thus, in order to apply Theorem 4.2 the sample complexity becomes $\widetilde{O}(N^2 \log(1/\delta)/[(1-\gamma)^3\varepsilon^2\rho^2])$. □

Next, we describe a TV indistinguishable algorithm that enjoys similar sample complexity guarantees. Similarly as before, we use the main result of Singh and Yee [1994] which shows that if $\|\widehat{Q} - Q^\star\|_\infty \le \varepsilon$, then the greedy policy with respect to $\widehat{Q}$, i.e., $\forall s \in \mathcal{S}, \widehat{\pi}(s) := \arg\max_{a \in \mathcal{A}^s} \widehat{Q}(s, a)$, is $\varepsilon/(1-\gamma)$-approximately optimal (cf. Theorem D.3). Thus, if we want to obtain an $\varepsilon$-approximately optimal policy, it suffices to obtain a $(1-\gamma)\varepsilon$-approximately optimal $Q$-function. The indistinguishable guarantee follows from the data-processing inequality This is formalized in Corollary 4.4.

**Corollary 4.4.** *Let $\varepsilon, \rho \in (0,1)^2$ and $\delta \in (0, \rho/5)$. There is a $\rho$-TV indistinguishable algorithm that outputs an $\varepsilon$-optimal policy with probability at least $1 - \delta$. Moreover, it has time and sample complexity $\widetilde{O}(N^2 \log(1/\delta)/[(1-\gamma)^5\varepsilon^2\rho^2])$.*

### 4.3   From TV Indistinguishability to Replicability

We now describe how we can transform the TV indistinguishable algorithms we provided to replicable ones. As we alluded to before, this transformation does not hurt the sample complexity, but requires exponential time in the state-action space. Our transformation is based on the approach proposed by Kalavasis et al. [2023] which holds when the input domain is *countable*. Its main idea is that when two random variables follow distributions that are $\rho$-close in TV-distance, then there is a way to couple them using only *shared randomness*. The implementation of this coupling is based on the *Poisson point process* and can be thought of a generalization of von Neumann's rejection-based sampling to handle more general domains. We underline that in general spaces without structure it is not known yet how to obtain such a coupling. However, even though the input domain of the Gaussian mechanism is *uncountable* and the result of Kalavasis et al. [2023] does not apply directly in our setting, we are able to obtain a similar transformation as they did. The main step required to perform this transformation is to find a reference measure with respect to which the algorithm is *absolutely continuous*. We provide these crucial measure-theoretic definitions below.

**Definition 4.5** (Absolute Continuity). Consider two measures $P, \mathcal{P}$ on a $\sigma$-algebra $\mathcal{B}$ of subsets of $W$. We say that $P$ is absolutely continuous with respect to $\mathcal{P}$ if for any $E \in \mathcal{B}$ such that $\mathcal{P}(E) = 0$, it holds that $P(E) = 0$.

Recall that $\mathscr{A}(S)$ denotes the *distribution* over outputs, when the input to the algorithm is $S$.

**Definition 4.6.** Given a learning rule $\mathscr{A}$ and reference probability measure $\mathcal{P}$, we say that $\mathscr{A}$ is absolutely continuous with respect to $\mathcal{P}$ if for any input $S$, $\mathscr{A}(S)$ is absolutely continuous with respect to $\mathcal{P}$.

We emphasize that this property should hold for every fixed sample $S$, i.e., the randomness of the samples are not taken into account.

We now define what it means for two learning rules to be *equivalent*.

**Definition 4.7** (Equivalent Learning Rules). Two learning rules $\mathscr{A}, \mathscr{A}'$ are *equivalent* if for every fixed sample $S$, it holds that $\mathscr{A}(S) \stackrel{d}{=} \mathscr{A}'(S)$, i.e., for the same input they induce the same distribution over outputs.

Using a coupling technique based on the Poisson point process, we can convert the TV indistinguishable learning algorithms we have proposed so far to equivalent ones that are replicable. See Algorithm A.2 for a description of how to output a sample from this coupling. Let us view $\mathscr{A}(S;r), \mathscr{A}(S';r)$ as random vectors with small TV distance. The idea is to implement the shared internal randomness $r$ using rejection sampling so that the "accepted" sample will be the same across two executions with high probability. For some background regarding the Poisson point process and the technical tools we use, we refer the reader to Appendix C.2.

Importantly, for every $S$, the output $\mathscr{A}(S)$ of the algorithms we have proposed in Section 4.1 and Section 4.2 follow a Gaussian distribution, which is absolutely continuous with respect to the Lebesgue

measure. Furthermore, the Lebesgue measure is $\sigma$-finite so we can use the coupling algorithm (cf. Algorithm A.2) of Angel and Spinka [2019], whose guarantees are stated in Theorem C.4. We are now ready to state the result regarding the improved $\rho$-replicable SQ oracle for multiple queries. Its proof is an adaptation of the main result of Kalavasis et al. [2023].

**Theorem 4.8** (Replicable SQ Oracle for Multiple Queries). *Let $\varepsilon, \rho \in (0,1)^2$ and $\delta \in (0, \rho/5)$. Let $\phi_1, \ldots, \phi_d$ be $d$ statistical queries with co-domain $[0,1]$. Assume that we can simultaneously estimate the true values of all $\phi_i$'s with accuracy $\varepsilon$ and confidence $\delta$ using $n(\varepsilon, \delta)$ total samples. Then, there exists a $\rho$-replicable algorithm that requires at most $n\left(\varepsilon\rho/[4\sqrt{8d \cdot \log(4d/\delta)}], \delta/2\right)$ many samples to output estimates $\widehat{v}_1, \ldots, \widehat{v}_d$ of the true values $v_1, \ldots, v_d$ with the guarantee that $\max_{i \in [d]} |\widehat{v}_i - v_i| \leq \varepsilon$, with probability at least $1 - \delta$.*

By using an identical argument, we can obtain $\rho$-replicable algorithms for $Q$-function estimation and policy estimation. Recall that $N$ is the number of state-action pairs of the MDP.

**Theorem 4.9.** *Let $\varepsilon, \rho \in (0,1)^2$ and $\delta \in (0, \rho/4)$. There is a $\rho$-replicable algorithm that outputs an $\varepsilon$-optimal $Q$-function with probability at least $1 - \delta$. Moreover, it has sample complexity $\widetilde{O}(N^2 \log(1/\delta)/[(1-\gamma)^3 \varepsilon^2 \rho^2])$.*

**Corollary 4.10.** *Let $\varepsilon, \rho \in (0,1)^2$ and $\delta \in (0, \rho/4)$. There is a $\rho$-replicable algorithm that outputs an $\varepsilon$-optimal policy with probability at least $1 - \delta$. Moreover, it has sample complexity $\widetilde{O}(N^2 \log(1/\delta)/[(1-\gamma)^5 \varepsilon^2 \rho^2])$.*

In Remark E.2 we explain why we cannot use a coordinate-wise coupling.

# 5 Approximately Replicable Policy Estimation

The definitions of replicability (cf. Definition 2.10, Definition 4.1) we have discussed so far suffer from a significant sample complexity blow-up in terms of the cardinality of the state-action space which can be prohibitive in many settings of interest. In this section, we propose *approximate replicability*, a relaxation of these definitions, and show that this property can be achieved with a significantly milder sample complexity compared to (exact) replicability. Moreover, this definition does not require shared internal randomness across the executions of the algorithm.

First, we define a general notion of *approximate* replicability as follows.

**Definition 5.1** (Approximate Replicability). *Let $\mathcal{X}, \mathcal{Y}$ be the input and output domains, respectively. Let $\kappa : \mathcal{Y} \times \mathcal{Y} \to \mathbb{R}_{\geq 0}$ be some distance function on $\mathcal{Y}$ and let $\rho_1, \rho_2 \in (0,1)^2$. We say that an algorithm $\mathscr{A}$ is $(\rho_1, \rho_2)$-approximately replicable with respect to $\kappa$ if for any distribution $\mathcal{D}$ over $\mathcal{X}$ it holds that $\mathbb{P}_{S, S' \sim \mathcal{D}^n, r, r' \sim \mathcal{R}}\{\kappa(\mathscr{A}(S; r), \mathscr{A}(S'; r')) \geq \rho_1\} \leq \rho_2$.*

In words, this relaxed version of Definition 2.8 requires that the outputs of the algorithm, when executed on two sets of i.i.d. data, using *independent* internal randomness across the two executions, are close under some appropriate distance measure. In the context of our work, the output of the learning algorithm is some policy $\pi : \mathcal{S} \to \Delta(\mathcal{A})$, where $\Delta(\mathcal{A})$ denotes the probability simplex over $\mathcal{A}$. Thus, it is natural to instantiate $\kappa$ as some *dissimilarity measure* of distributions like the total variation (TV) distance or the Renyi divergence. For the exact definition of these dissimilarity measures, we refer the reader to Appendix B. We now state the definition of an approximately replicable policy estimator.

**Definition 5.2** (Approximately Replicable Policy Estimator). *Let $\mathscr{A}$ be an algorithm that takes as input samples of state-action pair transitions and returns a policy $\pi$. Let $\kappa$ be some dissimilarity measure on $\Delta(\mathcal{A})$ and let $\rho_1, \rho_2 \in (0,1)^2$. We say that $\mathscr{A}$ is $(\rho_1, \rho_2)$-approximately replicable if for any MDP $M$ it holds that $\mathbb{P}_{S, S' \sim G, r, r' \sim \mathcal{R}}\{\max_{s \in \mathcal{S}} \kappa(\pi(s), \pi'(s)) \geq \rho_1\} \leq \rho_2$, where $G$ is the generator of state-action pair transitions, $\mathcal{R}$ is the source of internal randomness of $\mathscr{A}$, $\pi$ is the output of $\mathscr{A}$ on input $S, r$, and $\pi'$ is its output on input $S', r'$.*

To the best of our knowledge, the RL algorithms that have been developed for the model we are studying do not satisfy this property. Nevertheless, many of them compute an estimate $Q$ with the promise that $\|Q - Q^\star\|_\infty \leq \varepsilon$ [Sidford et al., 2018a, Agarwal et al., 2020, Li et al., 2020]. Thus, it is not hard to see that if we run the algorithm twice on independent data with independent internal randomness we have that $\|Q - Q'\|_\infty \leq 2\varepsilon$. This is exactly the main property that we need in order

to obtain approximately replicable policy estimators. The key idea is that instead of outputting the greedy policy with respect to this $Q$-function, we output a policy given by some *soft-max* rule. Such a rule is a mapping $\mathbb{R}_{\geq 0}^{\mathcal{A}} \to \Delta(\mathcal{A})$ that achieves two desiderata: (i) The distribution over the actions is "stable" with respect to perturbations of the $Q$-function. (ii) For every $s \in \mathcal{S}$, the value of the policy $V^{\pi}(s)$ that is induced by this mapping is "close" to $V^{\star}(s)$.

Formally, the stability of the soft-max rule is captured through its Lipschitz constant (cf. Definition B.3). In this setting, this means that whenever the two functions $Q, Q'$ are close under some distance measure (e.g. the $\ell_{\infty}$ norm), then the policies that are induced by the soft-max rule are close under some (potentially different) dissimilarity measure. The approximation guarantees of the soft-max rules are captured by the following definition.

**Definition 5.3** (Soft-Max Approximation; [Epasto et al., 2020]). Let $\varepsilon > 0$. A soft-max function $f : \mathbb{R}_{\geq 0}^{\mathcal{A}} \to \Delta(\mathcal{A})$ is *$\varepsilon$-approximate* if for all $x \in \mathbb{R}^{\mathcal{A}}$, $\langle f(x), x \rangle \geq \max_{a \in \mathcal{A}} x_a - \varepsilon$.

In this work, we focus on the soft-max rule that is induced by the exponential function (ExpSoftMax), which has been studied in several application domains [Gibbs, 1902, McSherry and Talwar, 2007, Huang and Kannan, 2012, Dwork et al., 2014, Gao and Pavel, 2017]. Recall $\pi(s, a)$ denotes the probability mass that policy $\pi$ puts on action $a \in \mathcal{A}^s$ in state $s \in \mathcal{S}$. Given some $\lambda > 0$ and $Q(s, a) \in \mathbb{R}^{\mathcal{S} \times \mathcal{A}}$, the induced randomized policy $\pi$ is given by $\pi(s, a) = \frac{\exp\{\lambda Q(s,a)\}}{\sum_{a' \in \mathcal{A}^s} \exp\{\lambda Q(s,a')\}}$. For a discussion about the advantages of using more complicated soft-max rules like the one developed in Epasto et al. [2020], we refer the reader to Appendix F.1.

We now describe our results when we consider approximate replicability with respect to the Renyi divergence and the Total Variation (TV) distance. At a high level, our approach is divided into two steps: 1) Run some $Q$-learning algorithm (e.g. [Sidford et al., 2018a, Agarwal et al., 2020, Li et al., 2020]) to estimate some $\widehat{Q}$ such that $\|Q^{\star} - \widehat{Q}\|_{\infty} \leq \varepsilon$. 2) Estimate the policy using some soft-max rule. One advantage of this approach is that it allows for flexibility and different implementations of these steps that better suit the application domain. An important lemma we use is the following.

**Lemma 5.4** (Exponential Soft-Max Approximation Guarantee; [McSherry and Talwar, 2007]). *Let $\varepsilon \in (0, 1), \alpha, p \geq 1$, and set $\lambda = \log(d)/\varepsilon$, where $d$ is the ambient dimension of the input domain. Then, $\mathrm{ExpSoftMax}$ with parameter $\lambda$ is $\varepsilon$-approximate and $2\lambda$-Lipschitz continuous (cf. Definition B.3) with respect to $(\ell_p, D_{\alpha})$, where $D_{\alpha}$ is the Renyi divergence of order $\alpha$.*

This is an important building block of our proof. However, it is not sufficient on its own in order to bound the gap of the $\mathrm{ExpSoftMax}$ policy and the optimal one. This is handled in the next lemma whose proof is postponed to Appendix F. Essentially, it can be viewed as an extension of the result in Singh and Yee [1994] to handle the soft-max policy instead of the greedy one.

**Lemma 5.5** (Soft-Max Policy vs Optimal Policy). *Let $\varepsilon_1, \varepsilon_2 \in (0, 1)^2$. Let $\widehat{Q} \in \mathbb{R}^{\mathcal{S} \times \mathcal{A}}$ be such that $\|\widehat{Q} - Q^{\star}\| \leq \varepsilon_1$. Let $\hat{\pi}$ be the $\mathrm{ExpSoftMax}$ policy with respect to $\widehat{Q}$ using parameter $\lambda = \log|\mathcal{A}|/\varepsilon_2$. Then, $\|V^{\hat{\pi}} - V^{\star}\|_{\infty} \leq (2\varepsilon_1 + \varepsilon_2)/(1-\gamma)$.*

Combining Lemma 5.4 and Lemma 5.5 yields the desired approximate replicability guarantees we seek. The formal proof of the following result is postponed to Appendix F. Recall we write $N := \sum_{s \in \mathcal{S}} |\mathcal{A}^s|$ to denote the total number of state-action pairs.

**Theorem 5.6.** *Let $\alpha \geq 1, \gamma, \delta, \rho_1, \rho_2 \in (0, 1)^4$, and $\varepsilon \in \left(0, (1-\gamma)^{-1/2}\right)$. There is a $(\rho_1, \rho_2)$-approximately replicable algorithm $\mathscr{A}$ with respect to the Renyi divergence $D_{\alpha}$ such that given access to a generator $G$ for any MDP $M$, it outputs a policy $\hat{\pi}$ for which $\|V^{\hat{\pi}} - V^{\star}\|_{\infty} \leq \varepsilon$ with probability at least $1 - \delta$. Moreover, $\mathscr{A}$ has time and sample complexity $\widetilde{O}(N \log(1/\min\{\delta, \rho_2\})/[(1-\gamma)^5 \varepsilon^2 \rho_1^2])$.*

## 6 Conclusion

In this work, we establish sample complexity bounds for a several notions of replicability in the context of RL. We give an extensive comparison of the guarantees under these different notions in Appendix G. We believe that our work can open several directions for future research. One immediate next step would be to verify our lower bound conjecture for replicable estimation of multiple independent coins (cf. Conjecture D.8). Moreover, it would be very interesting to extend our results to different RL settings, e.g. offline RL with linear MDPs, offline RL with finite horizon, and online RL.

## Acknowledgments

Amin Karbasi acknowledges funding in direct support of this work from NSF (IIS-1845032), ONR (N00014-19-1-2406), and the AI Institute for Learning-Enabled Optimization at Scale (TILOS). Lin Yang is supported in part by NSF Award 2221871 and an Amazon Faculty Award. Grigoris Velegkas is supported by TILOS, the Onassis Foundation, and the Bodossaki Foundation. Felix Zhou is supported by TILOS. The authors would also like to thank Yuval Dagan for an insightful discussion regarding the Gaussian mechanism.

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

# A  Omitted Algorithms

---

**Algorithm A.1** TV Indistinguishable Oracle for Multiple Query Estimation

---

1: $\widehat{\mu} = (\widehat{\mu_1}, \ldots, \widehat{\mu_d}) \leftarrow$ StatisticalQueryOracles $\left( \frac{\varepsilon\rho}{2\sqrt{8d \cdot \log(4d/\delta)}}, \frac{\delta}{2} \right)$

2: Sample $\widehat{v} \sim \mathcal{N}(\widehat{\mu}, \varepsilon^2/(8 \cdot \log(4d/\delta)) \cdot I_d)$

3: Output $\widehat{v}$

---

**Algorithm A.2** Sampling from Pairwise Optimal Coupling; [Angel and Spinka, 2019]

---

1: **Input:** collection of random vectors $\mathcal{S} = \{X\}$ absolutely continuous with respect to a $\sigma$-finite measure $\mu$, with densities $f_X : \mathbb{R}^d \to \mathbb{R}$, some $X \in \mathcal{S}$

2: Let $\mathcal{R}$ denote the Poisson point process over $\mathbb{R}^d \times \mathbb{R}_+ \times \mathbb{R}_+$ with intensity $\mu \times \text{Leb} \times \text{Leb}$.

3: Sample $r := \{(x_i, y_i, t_i) : i \in \mathbb{N}\} \sim \mathcal{R}$.

4: Let $i^\star \leftarrow \text{argmin}_{i \in \mathbb{N}}\{t_i : f_S(x_i) > y_i\}$.

5: Output $x_{i^\star}$ as a sample for $X$.

---

# B  Omitted Definitions

In this section, we discuss some dissimilarity measures for probability distributions that we use in this work.

**Definition B.1** (Total Variation (TV) Distance)**.** Let $\Omega$ be a countable domain and $P, Q$ be probability distributions over $\Omega$. The total variation distance between $P, Q$, denoted by $d_{\text{TV}}(P, Q)$ is defined as

$$
\begin{aligned}
d_{\text{TV}}(P, Q) &= \sup_{A \in 2^\Omega} P(A) - Q(A) \\
&= ||P - Q||_1 \\
&= \inf_{(X,Y) \sim \Pi(P,Q)} \mathbb{P}\{X \neq Y\},
\end{aligned}
$$

where $\Pi(P, Q)$ is a *coupling* between $P, Q$.

In the above definition, a coupling is defined to be a joint probability distribution $(X, Y) \sim \Pi(P, Q)$ over the product space whose marginals distributions are $P, Q$, i.e., if we only view the individual components, then $X \sim P, Y \sim Q$.

**Definition B.2** (Renyi Divergence)**.** Let $\Omega$ be a countable domain and $P, Q$ be probability distributions over $\Omega$. For any $\alpha > 1$ the Renyi divergence of order $\alpha$ between $P, Q$, denoted by $D_\alpha(P \| Q)$ is defined as

$$
D_a(P \| Q) = \frac{1}{\alpha - 1} \log \left( \sum_{\omega \in \Omega} \frac{P(\omega)^\alpha}{Q(\omega)^{\alpha - 1}} \right).
$$

We note that the quantity above is undefined when $\alpha = 1$. However, we can take its limit and define $D_1(P \| Q) = \sum_{\omega \in \Omega} P(\omega) \log \frac{P(\omega)}{Q(\omega)}$, which recovers the definition of KL-divergence. Similarly, we can define $D_\infty(P \| Q) = \max_{\omega \in \Omega} \frac{P(\omega)}{Q(\omega)}$.

Another important definition is that of Lipschitz continuity.

**Definition B.3** (Lipschitz Continuity)**.** Let $\mathcal{X}, \mathcal{Y}$ be two domains, $f : \mathcal{X} \to \mathcal{Y}$ be some function, and $d_1 : \mathcal{X} \times \mathcal{X} \to \mathbb{R}_{\geq 0}$, $d_2 : \mathcal{Y} \times \mathcal{Y} \to \mathbb{R}_{\geq 0}$ be distance measures over $\mathcal{X}, \mathcal{Y}$, respectively. We say that $f$ is $L$-Lipschitz continuous with respect to $d_1, d_2$ if $\forall x_1, x_2 \in \mathcal{X}$ it holds that $d_2(f(x_1), f(x_2)) \leq L \cdot d_1(x_1, x_2)$.

## B.1  Local Randomness

Our algorithms in Section 3 satisfy a property which we call *locally random*. This roughly means that for every decision an algorithm makes based on external and internal randomness, the internal randomness is used once and discarded immediately after.

**Definition B.4** (Locally Random). *Let* $\mathscr{A} = (\mathscr{A}^{(1)}, \ldots, \mathscr{A}^{(N)}) : \mathcal{I}^n \to \mathbb{R}^N$ *be an* $n$-*sample randomized algorithm that takes as input elements from some domain* $\mathcal{I}$ *and maps them to* $\mathbb{R}^N$. *We say that* $\mathscr{A}$ *is locally random if:*

(i) *The* $i$-*th output component* $\mathscr{A}^{(i)}(\bar{S}; \bar{r}^{(i)})$ *is a function of all samples* $\bar{S}$ *but only its own internal random string* $\bar{r}^{(i)}$.

(ii) *The sources* $\bar{r}^{(i)}$ *of internal randomness are independent of each other and the external samples* $\bar{S}$.

We will see that by restricting ourselves to locally random algorithms, it is necessary and suffcient to incur a sample cost of $\tilde{\Theta}(N^3)$ for replicable $Q$-estimation. However, by relaxing this restriction and allowing for internal randomness that is correlated, we can achieve $\tilde{O}(N^2)$ sample complexity.

## C Omitted Preliminaries

### C.1 Replicability

Impagliazzo et al. [2022] introduced the definition of replicability and demonstrated the following basic replicable operation.

**Theorem C.1** (Replicable SQ-Oracle; [Impagliazzo et al., 2022])**.** *Let* $\varepsilon, \rho \in (0,1)$ *and* $\delta \in (0, \rho/3)$. *Suppose* $\phi$ *is a sample mean statistical query with co-domain* $[0,1]$. *There is a locally random* $\rho$-*replicable SQ oracle to estimate its true value with tolerance* $\varepsilon$ *and failure rate* $\delta$. *Moreover, the oracle has sample complexity*

$$\tilde{O}\left(\frac{1}{\varepsilon^2 \rho^2} \log \frac{1}{\delta}\right).$$

Esfandiari et al. [2023b] generalized the result above to multiple general queries with unbounded co-domain, assuming some regularity conditions on the queries.

**Theorem C.2** (Replicable Rounding; [Impagliazzo et al., 2022, Esfandiari et al., 2023b])**.** *Let* $\varepsilon, \rho \in (0,1)$ *and* $\delta \in (0, \rho/3)$. *Suppose we have a finite class of statistical queries* $g_1, \ldots, g_N$ *with true values* $G_1, \ldots, G_N$ *and sampling* $n$ *independent points from* $\mathcal{D}$ *ensures that*

$$\sum_{j=1}^N |g_j(x_1, \ldots, x_n) - G_j| \le \varepsilon' := \frac{\varepsilon(\rho - 2\delta)}{\rho + 1 - 2\delta}$$

*with probability at least* $1 - \delta$.

*There is a locally random* $\rho$-*replicable algorithm that outputs estimates* $\widehat{G}_j$ *such that*

$$|\widehat{G}_j - G_j| \le \varepsilon$$

*with probability at least* $1 - \delta$ *for every* $j \in [N]$. *Moreover, it requires at most* $n$ *samples.*

### C.2 Coupling and Correlated Sampling

Our exposition in this section follows Kalavasis et al. [2023]. Coupling is a fundamental notion in probability theory with many applications [Levin and Peres, 2017]. The correlated sampling problem, which has applications in various domains, e.g., in sketching and approximation algorithms [Broder, 1997, Charikar, 2002], is described in Bavarian et al. [2016] as follows: Alice and Bob are assigned probability distributions $P_A$ and $P_B$, respectively, over a finite set $W$. *Without any communication, using only shared randomness* as the means to coordinate, Alice is required to output an element $x$ distributed according to $P_A$ and Bob is required to output an element $y$ distributed according to $P_B$. Their goal is to minimize the disagreement probability $\mathbb{P}\{x \neq y\}$, which is comparable with $d_{\text{TV}}(P_A, P_B)$. Formally, a correlated sampling strategy for a finite set $W$ with error $\varepsilon : [0,1] \to [0,1]$ is specified by a probability space $\mathcal{R}$ and a pair of functions $f, g : \Delta_W \times \mathcal{R} \to W$, which are measurable in their second argument, such that for any pair $P_A, P_B \in \Delta_W$ with $d_{\text{TV}}(P_A, P_B) \le \delta$, it holds that (i) the push-forward measure $f(P_A, \cdot)$ (resp. $g(P_B, \cdot)$) is $P_A$ (resp. $P_B$) and (ii) $\mathbb{P}_{r \sim \mathcal{R}}\{f(P_A, r) \neq g(P_B, r)\} \le \varepsilon(\delta)$. We underline that a correlated

sampling strategy is *not* the same as a coupling, in the sense that the latter requires a single function $h : \Delta_W \times \Delta_W \to \Delta_{W \times W}$ such that for any $P_A, P_B$, the marginals of $h(P_A, P_B)$ are $P_A$ and $P_B$ respectively.

It is known that for any coupling function $h$, it holds that $\mathbb{P}_{(x,y) \sim h(P_A, P_B)} \{x \neq y\} \geq d_{\mathrm{TV}}(P_A, P_B)$ and that this bound is attainable. Since $(f(P_A, \cdot), g(P_B, \cdot))$ induces a coupling, it holds that $\varepsilon(\delta) \geq \delta$ and, perhaps surprisingly, there exists a strategy with $\varepsilon(\delta) \leq \frac{2\delta}{1+\delta}$ [Broder, 1997, Kleinberg and Tardos, 2002, Holenstein, 2007] and this result is tight [Bavarian et al., 2016]. A second difference between coupling and correlated sampling has to do with the size of $W$: while correlated sampling strategies can be extended to infinite spaces $W$, it remains open whether there exists a correlated sampling strategy for general measure spaces $(W, \mathcal{F}, \mu)$ with any non-trivial error bound [Bavarian et al., 2016]. On the other hand, coupling applies to spaces $W$ of any size. For further comparisons between coupling and the correlated sampling problem of Bavarian et al. [2016], we refer to the discussion in Angel and Spinka [2019] after Corollary 4.

**Definition C.3** (Coupling). A coupling of two probability distributions $P_A$ and $P_B$ is a pair of random variables $(X, Y)$, defined on the same probability space, such that the marginal distribution of $X$ is $P_A$ and the marginal distribution of $Y$ is $P_B$.

A very useful tool for our derivations is a coupling protocol that can be found in Angel and Spinka [2019].

**Theorem C.4** (Pairwise Optimal Coupling [Angel and Spinka, 2019]). *Let $\mathcal{S}$ be any collection of random variables that are absolutely continuous with respect to a $\sigma$-finite measure $\mu$. Then, there exists a coupling of the variables in $\mathcal{S}$ such that for any $X, Y \in \mathcal{S}$,*

$$\mathbb{P}\{X \neq Y\} \leq \frac{2 d_{\mathrm{TV}}(X, Y)}{1 + d_{\mathrm{TV}}(X, Y)}.$$

*Moreover, this coupling requires sample access to a Poisson point process with intensity $\mu \times \mathrm{Leb} \times \mathrm{Leb}$, where $\mathrm{Leb}$ is the Lebesgue measure over $\mathbb{R}_+$, and full access to the densities of all the random variables in $\mathcal{S}$ with respect to $\mu$. Finally, we can sample from this coupling using Algorithm A.2.*

# D   Omitted Details from Section 3

Here we fill in the details from Section 3, which describes upper and lower bounds for replicable $Q$-function and policy estimation. In Appendix D.1, we provide the rigorous analysis of an algorithm for locally random replicable $Q$-function estimation as well as replicable policy estimation. Appendix D.3 describes the locall random replicable version of the multiple coin estimation problem, an elementary statistical problem that serves as the basis of our hardness proofs. Next, Appendix D.4 reduces locally random replicable $Q$-function estimation to the multiple coin estimation problem. Finally, Appendix D.5 reduces deterministic replicable policy estimation to locally random replicable $Q$-function estimation.

## D.1   Omitted Details from Upper Bounds

First, we show that we can use any non-replicable $Q$-function estimation algorithm as a black-box to obtain a locally random replicable $Q$-function estimation algorithm.

**Lemma D.1.** *Let $\varepsilon', \delta' \in (0, 1)$. Suppose there is an oracle $\mathcal{O}$ that takes $\widetilde{O}(f(N, \varepsilon', \delta'))$ samples from the generative model and outputs $\widehat{Q}$ satisfying $\|\widehat{Q} - Q^\star\|_\infty \leq \varepsilon'$ with probability at least $1 - \delta'$.*

*Let $\varepsilon, \rho \in (0, 1)$ and $\delta \in (0, \rho/3)$. There is a locally random $\rho$-replicable algorithm which makes a single call to $\mathcal{O}$ and outputs some $\bar{Q}$ satisfying $\|\bar{Q} - Q^\star\|_\infty \leq \varepsilon$ with probability at least $1 - \delta$. Moreover, it has sample complexity $\widetilde{O}\left(f\left(N, \varepsilon_0/N, \delta\right)\right)$ where $\varepsilon_0 := \varepsilon(\rho - 2\delta)/(\rho + 1 - 2\delta)$.*

*Proof (Lemma D.1).* Consider calling $\mathcal{O}$ with $\widetilde{O}(f(N, \varepsilon_0/N, \delta))$ samples. This ensures that

$$\sum_{s,a} \left| \widehat{Q}(s, a) - Q^\star(s, a) \right| \leq \varepsilon_0$$

with probability at least $1 - \delta$. By Theorem C.2, there is a locally random $\rho$-replicable algorithm that makes a single call to $\mathscr{O}$ with the number of samples above and outputs $\bar{Q}(s, a)$ such that

$$\left\|\bar{Q} - Q^\star\right\|_\infty \leq \varepsilon$$

with a probability of success of at least $1 - \delta$. $\qquad \square$

Sidford et al. [2018a] showed a (non-replicable) $Q$ function estimation algorithm which has optimal (non-replicable) sample complexity up to logarithmic factors.

Recall we write $N := \sum_{s \in \mathcal{S}} |\mathcal{A}^s|$ to denote the total number of state-action pairs.

**Theorem D.2** ([Sidford et al., 2018a]). *Let $\varepsilon, \delta \in (0, 1)$, there is an algorithm that outputs an $\varepsilon$-optimal policy, $\varepsilon$-optimal value function, and $\varepsilon$-optimal $Q$-function with probability at least $1 - \delta$ for any MDP. Moreover, it has time and sample complexity*

$$\widetilde{O}\left(\frac{N}{(1-\gamma)^3 \varepsilon^2} \log \frac{1}{\delta}\right).$$

The proof of Theorem 3.1, which we repeat below for convenience, follows by combining Lemma D.1 and Theorem D.2.

**Theorem 3.1.** *Let $\varepsilon, \rho \in (0, 1)^2$ and $\delta \in (0, \rho/3)$. There is a locally random $\rho$-replicable algorithm that outputs an $\varepsilon$-optimal $Q$-function with probability at least $1 - \delta$. Moreover, it has time and sample complexity $\widetilde{O}(N^3 \log(1/\delta) / [(1 - \gamma)^3 \varepsilon^2 \rho^2])$.*

The following result of Singh and Yee [1994] relates the $Q$-function estimation error to the quality of the greedy policy with respect to the estimated $Q$-function.

**Theorem D.3** ([Singh and Yee, 1994]). *Let $\widehat{Q}$ be a $Q$-function such that $||\widehat{Q} - Q^\star||_\infty \leq \varepsilon$. Let $\pi(s) := \arg\max_{a \in \mathcal{S}} \widehat{Q}(s, a), \forall s \in \mathcal{S}$. Then,*

$$\|V^\pi - V^\star\|_\infty \leq \frac{\varepsilon}{1 - \gamma}.$$

Theorem D.3 enables us to prove Corollary 3.2, an upper bound on the sample complexity of replicably estimating an $\varepsilon$-optimal policy. We restate the corollary below for convenience.

**Corollary 3.2.** *Let $\varepsilon, \rho \in (0, 1)^2$ and $\delta \in (0, \rho/3)$. There is a locally random $\rho$-replicable algorithm that outputs an $\varepsilon$-optimal policy with probability at least $1 - \delta$. Moreover, it has time and sample complexity $\widetilde{O}(N^3 \log(1/\delta) / [(1 - \gamma)^5 \varepsilon^2 \rho^2])$.*

*Proof (Corollary 3.2).* Apply the $\rho$-replicable algorithm from Theorem 3.1 to yield an $\varepsilon_0$-optimal $Q$-function and output the greedy policy based on this function. Theorem D.3 guarantees that the greedy policy derived from an $\varepsilon_0$-optimal $Q$-function is $\varepsilon_0/(1-\gamma)$-optimal. Choosing $\varepsilon_0 := (1 - \gamma)\varepsilon$ yields the desired result.

The replicability follows from the fact that conditioned on the event that the two $Q$-functions across the two runs are the same, which happens with probability at least $1 - \rho$, the greedy policies with respect to the underlying $Q$-functions will also be the same[6]. $\qquad \square$

## D.2 Lower Bounds for Replicable $Q$-Function & Policy Estimation

We now move on to the lower bounds and our approaches to obtain them. First, we describe a sample complexity lower bound for locally random $\rho$-replicable algorithms that seek to estimate $Q^\star$. Then, we reduce policy estimation to $Q$-estimation. Since the dependence of the sample complexity on the confidence parameter $\delta$ of the upper bound is at most polylogarithmic, the main focus of the lower bound is on the dependence on the size of the state-action space $N$, the error parameter $\varepsilon$, the replicability parameter $\rho$, and the discount factor $\gamma$.

---

[6]assuming a consistent tie-breaking rule

### D.2.1 Intuition of the $Q$-Function Lower Bound

Our MDP construction that witnesses the lower bound relies on the sample complexity lower bound for locally random algorithms that replicably estimate the biases of *multiple independent* coins. Impagliazzo et al. [2022] showed that any $\rho$-replicable algorithm that estimates the bias of a *single coin* with accuracy $\varepsilon$ requires at least $\Omega(1/\rho^2\varepsilon^2)$ samples (cf. Theorem D.13). We generalize this result and derive a lower bound for any locally random $\rho$-replicable algorithm that estimates the biases of $N$ coins with accuracy $\varepsilon$ and constant probability of success. We discuss our approach in Appendix D.2.2.

Next, given some $\varepsilon, \rho, \gamma, N$, we design an MDP for which estimating an approximately optimal $Q$-function is at least as hard as estimating $N$ coins. The main technical challenge for this part of the proof is to establish the correct dependence on the parameter $\gamma$ since it is not directly related to the coin estimation problem. We elaborate on it in Remark D.7.

*Remark* D.4. Our construction, combined with the non-replicable version of the coin estimation problem, can be used to simplify the construction of the non-replicable $Q$-estimation lower bound from Gheshlaghi Azar et al. [2013].

### D.2.2 The Replicable Coin Estimation Problem

Formally, the estimation problem, without the replicability requirement, is defined as follows.

**Problem D.5** (Multiple Coin Problem). Fix $q, \varepsilon, \delta \in (0,1)^3$ such that $q - \varepsilon \in (1/2, 1)$. Given sample access to $N$ independent coins each with a bias of either $q$ or $q - \varepsilon$, determine the bias of every coin with confidence at least $1 - \delta$.

We now informally state our main result for the multiple coin estimation problem, which could be useful in deriving replicability lower bounds beyond the scope of our work. See Theorem D.16 for the formal statement. Intuitively, this result generalizes Theorem D.13 to multiple instances.

**Theorem D.6** (Informal). *Suppose $\mathscr{A}$ is a locally random $\rho$-replicable algorithm for the multiple coin problem with a constant probability of success. Then, the sample complexity of $\mathscr{A}$ is at least*

$$\Omega\left(\frac{N^3 q(1-q)}{\varepsilon^2 \rho^2}\right).$$

Recall Yao's min-max principle [Yao, 1977], which roughly states that the expected cost of a randomized algorithm on its worst-case input is at least as expensive as the expected cost of any deterministic algorithm on random inputs chosen from some distribution. It is not clear how to apply Yao's principle directly, but we take inspiration from its essence and reduce the task of reasoning about a randomized algorithm with shared internal randomness to reasoning about a deterministic one with an additional layer of external randomness on top of the random flips of the coins.

Consider now a deterministic algorithm $g$ for distinguishing the bias of a single coin where the input bias is chosen uniformly in $[q - \varepsilon, q]$. That is, we first choose $p \sim U[q - \varepsilon, q]$, then provide i.i.d. samples from $\mathrm{Be}(p)$ to $g$. We impose some boundary conditions: if $p = q - \varepsilon$, $g$ should output "-" with high probability and if $p = q$, the algorithm should output "+" with high probability. We show that the probability of $g$ outputting "+" varies smoothly with respect to the bias of the input coin. Thus, there is an interval $I \subseteq (q - \varepsilon, q)$ such that $g$ outputs "-" or "+" with almost equal probability and so the output of $g$ is inconsistent across two executions with constant probability when $p$ lands in this interval. By the choice of $p \sim U[q - \varepsilon, q]$, if $\ell(I)$ denotes the length of $I$, then the output of $g$ is inconsistent across two executions with probability at least $\Omega(\ell(I)/\varepsilon)$. Quantifying $\ell(I)$ and rearranging yields the lower bound for a single coin.

For the case of $N$ independent coins, we use the pigeonhole principle to reduce the argument to the case of a single coin. The formal statement and proof of Theorem D.6 is deferred to Appendix D.3.

*Remark* D.7. The lower bound from Impagliazzo et al. [2022] for the single-coin estimation problem holds for the regime $q, q - \varepsilon \in (1/4, 3/4)$. We remove this constraint by analyzing the dependence of the lower bound on $q$. When reducing $Q$-function estimation to the multiple coin problem, the restricted regime yields a lower bound proportional to $(1 - \gamma)^{-2}$. In order to derive the stronger lower bound of $(1 - \gamma)^{-3}$, we must be able to choose $q \approx \gamma$ which can be arbitrarily close to 1.

In Section 4, we show that allowing for non-locally random algorithms enables us to shave off a factor of $N$ in the sample complexity. We also conjecture that this upper bound is tight.

**Conjecture D.8.** Suppose $\mathscr{A}(\bar{c}^{(1)}, \ldots, \bar{c}^{(N)}; \bar{r})$ is a randomized $\rho$-replicable algorithm for the multiple coin problem and has a constant probability of success. Then, the sample complexity of $\mathscr{A}$ is at least

$$\Omega\left(\frac{N^2 q(1-q)}{\varepsilon^2 \rho^2}\right).$$

### D.2.3 A Lower Bound for Replicable $Q$-Function Estimation

We now present the MDP construction that achieves the desired sample complexity lower bound. We define a family of MDPs $\mathbb{M}$ as depicted in Figure D.1. This particular construction was first presented by Mannor and Tsitsiklis [2004] and generalized by Gheshlaghi Azar et al. [2013], Feng et al. [2019].

Any MDP $M \in \mathbb{M}$ is parameterized by positive integers $K_M, L_M$, and some $p_M^{(k,\ell)} \in [0,1]$ for $k \in [K_M], \ell \in [L_M]$. The state space of $M$ is the disjoint union[7] of three sets $\mathcal{S} = \mathcal{X} \sqcup \mathcal{Y} \sqcup \mathcal{Z}$, where $\mathcal{X}$ consists of $K$ states $\{x_1, \ldots, x_K\}$ and each of them has $L$ available actions $\{a_1, \ldots, a_L\} =: \mathcal{A}$. All states in $\mathcal{Y}, \mathcal{Z}$ have a single action that the agent can take. Remark that each $M \in \mathbb{M}$ has $N = \sum_{s \in S} |\mathcal{A}^s| = 4K_M L_M$.

For $x \in \mathcal{X}$, by taking action $a \in \mathcal{A}$, the agent transitions to a state $y(x,a) \in \mathcal{Y}$ with probability 1. Let $p_M(x_k, a_\ell) := p_M^{(k,\ell)}$. For state $y(x,a) \in \mathcal{Y}$, we transition back to $y(x,a)$ with probability $p_M(x,a)$ and to $z(x,a) \in \mathcal{Z}$ with probability $1 - p_M(x,a)$. Finally, the agent always returns to $z(x,a)$ for all $z(x,a) \in \mathcal{Z}$. The reward function $r_M(s,a) = 1$ if $s \in \mathcal{X} \cup \mathcal{Y}$ and is 0 otherwise. We remark that for every $x \in \mathcal{X}, a \in \mathcal{A}$, its $Q^\star$ function can be computed in closed form by solving the Bellman optimality equation

$$Q_M^\star(x,a) = 1 + \gamma\left[p_M(x,a) \cdot Q_M^\star(x,a) + (1 - p_M(x,a)) \cdot 0\right] = \frac{1}{1 - \gamma p_M(x,a)}.$$

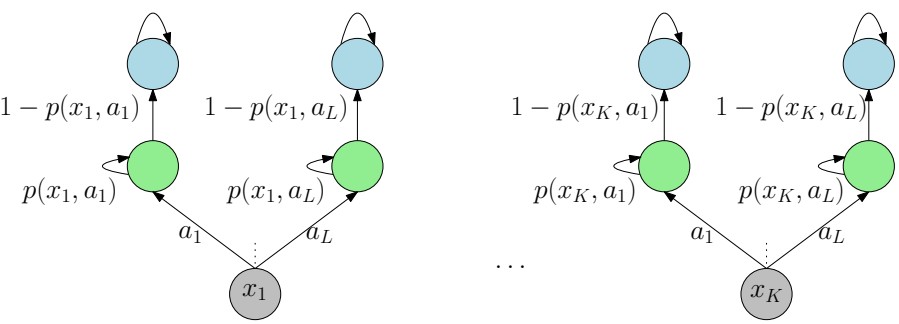

Figure D.1: The class of MDPs considered to prove the lower bound in Theorem D.9.

Recall we write $N := \sum_{s \in \mathcal{S}} |\mathcal{A}^s|$. to denote the total number of state-action pairs. Our main result in this section is the following.

**Theorem D.9.** *Let $\rho, \varepsilon \in (0,1)^2$, $\gamma \in (1/2, 1)$, and $\delta = 1/4$. Suppose $\mathscr{A}$ is a locally random $\rho$-replicable algorithm that returns an estimate $\widehat{Q}$ for any MDP with discount factor $\gamma$ such that $|\widehat{Q}(s,a) - Q^\star(s,a)| \leq \varepsilon$ with probability at least $1 - \delta/6$ for each $s \in \mathcal{S}, a \in \mathcal{A}^s$. Then $\mathscr{A}$ has a sample complexity of at least*

$$\Omega\left(\frac{N^3}{(1-\gamma)^3 \varepsilon^2 \rho^2}\right).$$

*Remark* D.10. If Conjecture D.8 holds, we obtain a sample complexity lower bound of

$$\Omega\left(\frac{N^2}{(1-\gamma)^3 \varepsilon^2 \rho^2}\right)$$

for general randomized $\rho$-replicable algorithms for $Q$ estimation.

---

[7]Denoted by $\sqcup$.

On a high level, we argue that a locally random $\rho$-replicable algorithm $\mathscr{A}$ for estimating the $Q$ function of arbitrary MDPs up to accuracy $\varepsilon \approx \varepsilon_0/(1-\gamma)^2$ yields a locally random $\rho$-replicable algorithm for the multiple coin problem (cf. Problem D.5) with tolerance approximately $\varepsilon_0 \approx (1-\gamma)^2 \varepsilon$ when we choose $q \approx \gamma$ in Theorem D.6. We can then directly apply Theorem D.6 to conclude the proof. See Appendix D.4 for details.

### D.2.4 A Lower Bound for Replicable Policy Estimation

Having established the lower bound for locally random replicable $Q$-function estimation, we now present our lower bound for deterministic replicable policy estimation. We argue that a deterministic $\rho$-replicable algorithm for optimal policy estimation yields a locally random $\rho$-replicable algorithm for optimal $Q$-function estimation after some post-processing that has sample complexity $\tilde{o}\left(N^3/\varepsilon^2\rho^2(1-\gamma)^3\right)$. It follows that the sample complexity lower bound we derived for $Q$-function estimation holds for policy estimation as well.

In order to describe the post-processing step, we employ a locally random replicable rounding algorithm (cf. Theorem C.2) that is provided in Esfandiari et al. [2023b]. Intuitively, we show that estimating the value function $V^\pi$ of $\pi$ reduces to estimating the optimal $Q$-function of some single-action MDP. Given such an estimate $\hat{V}^\pi$, we can then estimate $Q^\pi$ using the simple sample mean query given sufficient samples from the generative model. Lastly, the locally random replicable rounding subroutine from Theorem C.1 is used as a post-processing step.

We now state the formal lower bound regarding the sample complexity of deterministic replicable policy estimation. Its proof follows by combining the $Q$-function estimation lower bound and the reduction we described above. For the full proof, see Appendix D.5.

**Theorem D.11.** *Let $\varepsilon, \rho \in (0,1)^2$ and $\delta = 1/4$. Suppose $\mathscr{A}$ is a deterministic $\rho$-replicable algorithm that outputs a randomized policy $\pi$ such that $|V^\pi(s) - V^\star(s)| \le \varepsilon$ with probability at least $1 - \delta/12$ for each $s \in \mathcal{S}$. Then $\mathscr{A}$ has a sample complexity of at least*

$$\Omega\left(\frac{N^3}{(1-\gamma)^3\varepsilon^2\rho^2}\right).$$

*Remark* D.12. If Conjecture D.8 holds, we obtain a sample complexity lower bound of

$$\Omega\left(\frac{N^2}{(1-\gamma)^3\varepsilon^2\rho^2}\right)$$

for general randomized $\rho$-replicable algorithms for policy estimation.

### D.3 The Coin Problem

As mentioned before, estimating the bias of a coin is an elementary statistical problem. In order to establish our lower bounds, we first aim to understand the sample complexity of locally random algorithms that replicably estimate the bias of multiple coins simultaneously.

Impagliazzo et al. [2022] explored the single coin version of the problem and established a version of the following result when the biases $q, q - \varepsilon$ of the coins in question lie in the interval $(1/4, 3/4)$. We state and prove the more general result when we allow $q$ to be arbitrarily close to 1.

**Theorem D.13** (SQ-Replicability Lower Bound; [Impagliazzo et al., 2022]). *Fix $q, \varepsilon, \rho \in (0,1)^3$ such that $q - \varepsilon \in (1/2, 1)$ and $\delta = 1/4$. Let $\mathscr{A}(\bar{c}; \bar{r})$ be an algorithm for the (single) coin problem.*

*Suppose $\mathscr{A}$ satisfies the following:*

    (i) *$\mathscr{A}$ outputs $\{0,1\}$ where 1 indicates its guess that the bias of the coin which generated $\bar{c}$ is $q$.*

    (ii) *$\mathscr{A}$ is $\rho$-replicable even when its samples are drawn from coins with bias $p \in [q - \varepsilon, q]$ for all $i \in [N]$.*

    (iii) *If $p \in \{q - \varepsilon, q\}$, then $\mathscr{A}$ correctly guesses the bias of the $i$-th coin with probability at least $1 - \delta/6$.*

*Then the sample complexity of $\mathscr{A}$ is at least*

$$\Omega\left(\frac{q(1-q)}{\varepsilon^2\rho^2}\right).$$

We follow a similar proof process as Impagliazzo et al. [2022] towards a lower bound for multiple coins. In particular, we begin with the two following lemmas adapted from Impagliazzo et al. [2022].

**Lemma D.14** ([Impagliazzo et al., 2022]). *Let* $g : \{0,1\}^m \to \{0,1\}$ *be a boolean function. Suppose the input bits are independently sampled from* $\mathrm{Be}(p)$ *for some parameter* $p \in [0,1]$ *and let* $\mathrm{Acc} : [0,1] \to [0,1]$ *be given by*

$$\mathrm{Acc}(p) := \mathbb{P}_{\bar{x} \sim_{i.i.d.} \mathrm{Be}(p)}\{g(\bar{x}) = 1\}.$$

*Then* $\mathrm{Acc}$ *is differentiable on* $(0,1)$ *and for all* $p \in (0,1)$,

$$\mathrm{Acc}'(p) \leq \sqrt{\frac{m}{p(1-p)}}.$$

*Proof (Lemma D.14).* Fix $p \in (0,1)$ and suppose $\bar{x} \sim_{i.i.d.} \mathrm{Be}(p)$. Define

$$a_k := \mathbb{P}\left\{ g(\bar{x}) = 1 \,\middle|\, \sum_{i=1}^{m} x_i = k \right\}.$$

Then

$$\mathrm{Acc}(p) = \sum_{k=1}^{m} a_k \binom{m}{k} p^k (1-p)^{m-k}.$$

In particular, $\mathrm{Acc}$ is differentiable.

By computation,

$$
\begin{aligned}
\mathrm{Acc}'(p) &= \sum_{k=1}^{m} a_k \binom{m}{k} \left[ k p^{k-1}(1-p)^{m-k} - (m-k)p^k(1-p)^{m-k-1} \right] \\
&= \sum_k a_k \binom{m}{k} p^k (1-p)^{m-k} \left[ \frac{k}{p} - \frac{m-k}{1-p} \right] \\
&= \sum_k a_k \binom{m}{k} p^k (1-p)^{m-k} \cdot \frac{k(1-p) - (m-k)p}{p(1-p)} \\
&= \sum_k a_k \binom{m}{k} p^k (1-p)^{m-k} \cdot \frac{k - mp}{p(1-p)} \\
&\leq \sum_k \binom{m}{k} p^k (1-p)^{m-k} \cdot \frac{|k - mp|}{p(1-p)} && a_k \in [0,1] \\
&= \frac{1}{p(1-p)} \mathbb{E}\left[ |X - \mathbb{E}[X]| \right] && X \sim \mathrm{Bin}(m,p) \\
&\leq \frac{1}{p(1-p)} \sqrt{\mathrm{Var}[X]} \\
&= \frac{1}{p(1-p)} \sqrt{mp(1-p)} \\
&= \sqrt{\frac{m}{p(1-p)}}. && \square
\end{aligned}
$$

In the following arguments, we need to reason about probabilistic events with multiple sources of randomness. For the sake of clarity, we use the formal definitions of a probability space and random variable. Specifically, let $(W, \mathcal{F}, \mathbb{P})$ be an underlying probability space where $W$ is some sample space, $\mathcal{F} \subseteq 2^W$ is a $\sigma$-algebra, and $\mathbb{P}$ is a probability measure. A random variable is simply a real-valued function $W \to \mathbb{R}$ from the sample space.

Define $C_p := \mathrm{Be}(p)$. Moreover, define $C_- := \mathrm{Be}(q - \varepsilon)$ and $C_+ := \mathrm{Be}(q)$ to be the distributions of the possible biased coins.

**Lemma D.15** ([Impagliazzo et al., 2022]). *Fix* $q, \varepsilon \in (0,1)$ *such that* $q - \varepsilon \in (1/2, 1)$ *and* $\delta = 1/4$. *Suppose* $g : \{0,1\}^m \to \{0,1\}$ *is a boolean function satisfying the following:*

*(i) For $\bar{x} \sim_{i.i.d.} C_-$, $\mathbb{P}\{w : g(\bar{x}(w)) = 0\} \geq 1 - \delta$.*

*(ii) For $\bar{x} \sim_{i.i.d.} C_+$, $\mathbb{P}\{w : g(\bar{x}(w)) = 1\} \geq 1 - \delta$.*

*Then for $p \sim U[q - \varepsilon, q]$ and $\bar{x}^{(1)}, \bar{x}^{(2)} \sim_{i.i.d.} C_p$,*

$$\mathbb{P}\{w : g(\bar{x}^{(1)}(w)) \neq g(\bar{x}^{(2)}(w))\} \geq \Omega\left(\frac{\sqrt{q(1-q)}}{\varepsilon\sqrt{m}}\right).$$

*In other words, we require*

$$m \geq \Omega\left(\frac{q(1-q)}{\varepsilon^2 \rho^2}\right)$$

*if we wish to reduce the probability above to at most $\rho$.*

We should interpret the input of the function $g$ from Lemma D.15 as $m$ realizations of coin flips from the same coin and the output of $g$ as its guess whether the bias of the coin that generated the realizations is $q$. Lemma D.15 states that if the function $g$ is "nice", i.e. is able to distinguish the two coins $C_-, C_+$ with some fixed confidence $1 - \delta$, then the same function is not replicable with probability at least $\Omega(\sqrt{q(1-q)}/\varepsilon\sqrt{m})$ when the bias is chosen uniformly randomly in the interval $[q - \varepsilon, q]$. Let us view an arbitrary $\rho$-replicable algorithm $\mathscr{A}(\bar{c}; \bar{r})$ as a distribution over functions $g_{\bar{r}}(\bar{c})$. Impagliazzo et al. [2022] argued that at least a constant fraction of $g_{\bar{r}}$ is nice, leading to Theorem D.13. Unfortunately, this argument does not extend trivially to the case of multiple coins. However, we present an extension for the special case when $\mathscr{A}$ is locally random.

*Proof (Lemma D.15).* Let $g : \{0,1\}^m \to \{0,1\}$ be a boolean function that satisfies the condition of Lemma D.15. By Lemma D.14,

$$\text{Acc}(p) := \mathbb{P}_{\bar{c} \sim_{i.i.d.} C_p}\{w : g(\bar{c}(w)) = 1\}$$

is differentiable (continuous) and for every $p \in [q - \varepsilon, q]$,

$$\text{Acc}'(p) \leq \sqrt{\frac{m}{q(1-q)}}.$$

This is because $\frac{1}{p(1-p)}$ is non-increasing on $(1/2, 1)$. In particular, $\text{Acc}(p)$ is $\sqrt{\frac{m}{q(1-q)}}$-Lipschitz over the interval $[q - \varepsilon, q]$.

Now, $\text{Acc}(q - \varepsilon) \leq \delta < 1/4$ and $\text{Acc}(q) \geq 1 - \delta > 3/4$, thus by the intermediate value theorem from elementary calculus, there is some $q_0 \in (q - \varepsilon, q)$ such that $\text{Acc}(q_0) = 1/2$. It follows by the Lipschitz condition and the mean value theorem that there is an interval $I$ of length $\Omega(\sqrt{q(1-q)/m})$ around $q_0$ so that for all $p \in I$,

$$\text{Acc}(p) \in \left(\frac{1}{3}, \frac{2}{3}\right).$$

For two independent sequences of i.i.d. samples, say $\bar{x}^{(1)}, \bar{x}^{(2)}$, we know that $g(\bar{x}^{(1)})$ and $g(\bar{x}^{(2)})$ are independent. Thus for $p \in I$, there is a $2\,\text{Acc}(p)(1 - \text{Acc}(p)) > 4/9$ probability that $g(\bar{x}^{(1)}) \neq g(\bar{x}^{(2)})$.

Then for $p \sim U[q - \varepsilon, q]$ and $\bar{x}^{(1)}, \bar{x}^{(2)} \sim_{i.i.d.} C_p$,

$$\mathbb{P}\Big\{w : g(\bar{x}^{(1)}(w)) \neq g(\bar{x}^{(2)}(w))\Big\}$$

$$\geq \mathbb{P}\Big\{w : g(\bar{x}^{(1)}(w)) \neq g(\bar{x}^{(2)}(w)) \,\Big|\, p(w) \in I\Big\} \cdot \mathbb{P}\{w : p(w) \in I\}$$

$$\geq \frac{4}{9} \cdot \Omega\left(\frac{\sqrt{q(1-q)/m}}{2\varepsilon}\right) \qquad\qquad p \sim U[q - \varepsilon, q]$$

$$= \Omega\left(\frac{\sqrt{q(1-q)}}{\varepsilon\sqrt{m}}\right). \qquad\qquad \square$$

Lemma D.15 enables us to prove Theorem D.13.

*Proof (Theorem D.13).* Let $G_-$ be the event that $\mathscr{A}$ correctly guess the bias of the coin given $p = q - \varepsilon$ and similarly for $G_+$. Thus

$$G_- := \{w : \mathscr{A}(\bar{c}(w); \bar{r}(w)) = 0\} \qquad\qquad p = q - \varepsilon$$
$$G_+ := \{w : \mathscr{A}(\bar{c}(w); \bar{r}(w)) = 1\} \qquad\qquad p = q.$$

Here the randomness lies only in the samples $\bar{c} \sim_{i.i.d.} C_p$ and the uniformly random $\bar{r}$. By (iii),

$$\frac{\delta}{3 \cdot 2} \geq \sum_{\bar{r}} \mathbb{P}(G_-^c \mid \bar{r}) \cdot \mathbb{P}(\bar{r})$$
$$\geq \mathbb{E}_{\bar{r}}[\mathbb{P}(G_-^c \mid \bar{r})].$$

Thus Markov's inequality tells us that choosing $\bar{r}$ uniformly at random satisfies

$$\mathbb{P}(G_-^c \mid \bar{r}) \leq \delta$$

with probability at least $1 - 1/(3 \cdot 2)$ and similarly for $G_+$.

By a union bound, choosing $\bar{r}$ uniformly at random means $G_\pm$ both occur with probability at least $2/3$. Let us write $\bar{r} \in G := G_+ \cap G_-$ as a shorthand for indicating the random draw $\bar{r}$ satisfies both $G_\pm$.

Fix $\bar{r}^\star \in G$ to be any particular realization and observe that $g := \mathscr{A}(\,\cdot\,; \bar{r}^\star)$ satisfies the condition of Lemma D.15. By Lemma D.15, if $p \sim U[q - \varepsilon, q]$, and $\bar{c}^{(1)}, \bar{c}^{(2)} \sim_{i.i.d.} C_p$,

$$\mathbb{P}\Big\{w : \mathscr{A}(\bar{c}^{(1)}(w); r(w)) \neq \mathscr{A}(\bar{c}^{(2)}(w); r(w))\Big\}$$
$$= \sum_{\bar{r}} \mathbb{P}\Big\{w : \mathscr{A}(\bar{c}^{(1)}(w); r(w)) \neq \mathscr{A}(\bar{c}^{(2)}(w); r(w)) \mid \bar{r}\Big\} \cdot \mathbb{P}(\bar{r})$$
$$\geq \sum_{\bar{r} \in G} \mathbb{P}\Big\{w : \mathscr{A}(\bar{c}^{(1)}(w); r(w)) \neq \mathscr{A}(\bar{c}^{(2)}(w); r(w)) \mid \bar{r}\Big\} \cdot \mathbb{P}(\bar{r})$$
$$\geq \frac{2}{3} \cdot \Omega\left(\frac{\sqrt{q(1-q)}}{\varepsilon\sqrt{m}}\right)$$
$$= \Omega\left(\frac{\sqrt{q(1-q)}}{\varepsilon\sqrt{m}}\right).$$

We remark that in the derivation above, a crucial component is the fact that outputs of $\mathscr{A}$ across two runs are conditionally independent given the internal randomness.

Since (ii) also holds when $p$ is chosen uniformly at random in the interval $[q - \varepsilon, q]$, it follows that $\Omega(\sqrt{q(1-q)}/\varepsilon\sqrt{m}) \leq \rho$ is a lower bound for the replicability parameter. The total sample complexity is thus at least

$$\Omega\left(\frac{q(1-q)}{\varepsilon^2 \rho^2}\right). \qquad\qquad \square$$

We now use Theorem D.13 to prove the formal statement of Theorem D.6, a samples complexity lower bound for locally random algorithms that replicably distinguish the biases of $N$ independent coins $C^{(1)}, \ldots, C^{(N)}$, assuming each of which is either $C_-$ or $C_+$. The lower bound for locally random replicable $Q$-estimation follows. We formally state Theorem D.6 below.

**Theorem D.16** (Theorem D.6; Formal). *Fix $q, \varepsilon, \rho \in (0,1)^3$ such that $q - \varepsilon \in (1/2, 1)$ and $\delta = 1/4$. Let $\mathscr{A} = (\mathscr{A}^{(1)}, \ldots, \mathscr{A}^{(N)})$ be a locally random algorithm for the multiple coin problem where $\mathscr{A}^{(i)}(\bar{c}^{(1)}, \ldots, \bar{c}^{(N)}; \bar{r}^{(i)})$ is the output for the $i$-th coin that draws internal randomness independently from other coins.*

*Suppose $\mathscr{A}$ satisfies the following:*

*(i) $\mathscr{A}^{(i)}$ outputs $\{0,1\}$ where 1 indicates its guess that the bias of the coin which generated $\bar{c}^{(i)}$ is $q$.*

(ii) $\mathscr{A}$ is $\rho$-replicable even when its samples are drawn from coins where $p^{(i)} \in [q - \varepsilon, q]$ for all $i \in [N]$.

(iii) If $p^{(i)} \in \{q - \varepsilon, q\}$, then $\mathscr{A}$ correctly guesses the bias of the $i$-th coin with probability at least $1 - \delta/6$.

*Then the sample complexity of $\mathscr{A}$ is at least*

$$\Omega\left(\frac{N^3 q(1-q)}{\varepsilon^2 \rho^2}\right).$$

We remark that the scaling with respect to $q$ is important in order to establish a strong lower bound with respect to the parameter $(1 - \gamma)^{-1}$.

*Proof (Theorem D.16).* First, we remark that the $i$-th output of $\mathscr{A}$ across two runs is independent of other outputs since both the coin flips and internal randomness are drawn independently per coin. Thus we may as well assume that $\mathscr{A}^{(i)}$ only reads the coin flips from the $i$-th coin.

Let $\rho_i, i \in [N]$ be the probability that the output of $\mathscr{A}^{(i)}$ is inconsistent across two executions when the bias $p^{(i)}$ is chosen uniformly in $[q - \varepsilon, q]$. We claim that there are at least $N/2$ indices $i \in [N]$ such that $\rho_i \leq \rho/N$. Indeed, by the independence due to the local randomness property, we have

$$\rho \geq 1 - \prod_{i \in [N]} (1 - \rho_i)$$

$$\geq 1 - \exp\left(-\sum_i \rho_i\right) \qquad\qquad 1 + x \leq e^x$$

$$\geq \sum_i \frac{\rho_i}{2}. \qquad\qquad e^{-x} \leq 1 - \frac{x}{2}, x \in [0, 1]$$

Suppose towards a contradiction that more than $N/2$ indices $i \in [N]$ satisfy $\rho_i > \rho/N$. But then

$$\sum_i \frac{\rho_i}{2} > \frac{N}{2} \cdot 2\frac{\rho}{N} = \rho,$$

which is a contradiction.

Let $I \subseteq [N]$ be the indices of the coins such that $\rho_i \leq \rho/N$. For each $i \in I$, $\mathscr{A}^{(i)}$ satisfies the conditions for Theorem D.13. Thus $\mathscr{A}^{(i)}$ has sample complexity at least

$$\Omega\left(\frac{q(1-q)}{\varepsilon^2 (\rho/N)^2}\right).$$

It follows that the total sample complexity is at least

$$\Omega\left(\frac{N}{2} \cdot \frac{q(1-q)}{\varepsilon^2 (\rho/N)^2}\right) = \Omega\left(\frac{N^3 q(1-q)}{\varepsilon^2 \rho^2}\right). \qquad\qquad \square$$

### D.4 Replicable $Q$-Function Estimation Lower Bound

In this section, we restate and prove Theorem D.9, a lower bound on locally random replicable $Q$-function estimation.

**Theorem D.9.** *Let $\rho, \varepsilon \in (0, 1)^2$, $\gamma \in (1/2, 1)$, and $\delta = 1/4$. Suppose $\mathscr{A}$ is a locally random $\rho$-replicable algorithm that returns an estimate $\widehat{Q}$ for any MDP with discount factor $\gamma$ such that $|\widehat{Q}(s, a) - Q^\star(s, a)| \leq \varepsilon$ with probability at least $1 - \delta/6$ for each $s \in \mathcal{S}, a \in \mathcal{A}^s$. Then $\mathscr{A}$ has a sample complexity of at least*

$$\Omega\left(\frac{N^3}{(1-\gamma)^3 \varepsilon^2 \rho^2}\right).$$

*Proof (Theorem D.9).* Choose $q := {}^{4\gamma - 1}/3\gamma$ and $\varepsilon_0 \in (0, {}^{(1-\gamma)}/\gamma)$ such that $q - \varepsilon_0 \in ({}^1/2, 1)$. Let $C^{(i)} \sim \mathrm{Be}(p^{(i)})$ be Bernoulli random variables (biased coins) for $i \in [N]$ where $p^{(i)} \in [q - \varepsilon_0, q]$.

To see the reduction, first choose any $K, L \in \mathbb{Z}_+$ such that $KL = N$ and let $M \in \mathbb{M}$ be such that the state action space has cardinality $4N$ as in the construction in Figure D.1. We identity each $i \in [N]$ with a pair $(x, a) \in \mathcal{X} \times \mathcal{A}$ and write $i_{x,a}$ to be the index corresponding to the pair $(x, a)$, i.e. $p(x, a) = p^{(i_{x,a})}$. For each state $y(x, a) \in \mathcal{Y}$, we flip the coin $C^{(i_{x,a})} \sim \mathrm{Be}(p(x, a)))$ and transition back to $y(x, a)$ if the coin lands on 1 and to $z(x, a) \in \mathcal{Z}$ if it lands on 0. By construction,

$$Q_M^\star(x, a) = \frac{1}{1 - \gamma p_M(x, a)}.$$

Let us compare the absolute difference of $Q^\star(x, a)$ when $p(x, a) = q$ versus when $p(x, a) = q - \varepsilon_0$. By computation,

$$\frac{1}{1 - \gamma q} - \frac{1}{1 - \gamma(q - \varepsilon_0)} = \frac{\gamma \varepsilon_0}{[1 - \gamma(q - \varepsilon_0)][1 - \gamma q]}.$$

Then the following inequalities hold.

$$q \geq \frac{2}{3}$$

$$1 - q = \frac{1 - \gamma}{3\gamma}$$

$$\geq \frac{2}{3}(1 - \gamma) \qquad\qquad \gamma > \frac{1}{2}$$

$$1 - \gamma q = \frac{4}{3}(1 - \gamma)$$

$$1 - \gamma(q - \varepsilon_0) = \frac{4}{3}(1 - \gamma) + \gamma \varepsilon_0$$

$$\leq \frac{4}{3}(1 - \gamma) + (1 - \gamma) \qquad\qquad \varepsilon_0 < \frac{1 - \gamma}{\gamma}$$

$$= \frac{7}{3}(1 - \gamma).$$

It follows that

$$\left| \frac{1}{1 - \gamma q} - \frac{1}{1 - \gamma(q - \varepsilon_0)} \right| \geq \frac{9\gamma \varepsilon_0}{28(1 - \gamma)^2}$$

$$\geq \frac{3\varepsilon_0}{14(1 - \gamma)^2} \qquad\qquad \gamma \geq \frac{2}{3}$$

$$=: \varepsilon.$$

Suppose we run a locally random algorithm that replicably estimates $Q_M^\star(x, a)$ to an accuracy of ${}^\varepsilon/3$. Then we are able to distinguish whether the biases of the coins $C^{(i_{x,a})}$ is either $q - \varepsilon_0$ or $q$. In addition, the internal randomness consumed in the estimation of $C^{(i_{x,a})}$ comes only from the internal randomness used to estimate $Q^\star(x, a)$. Hence the scheme we described is a locally random algorithm for the multiple coin problem. Finally, the scheme is replicable even when the biases are chosen in the interval $[q - \varepsilon_0, q]$.

By Theorem D.16, the following sample complexity lower bound holds for $\mathscr{A}$:

$$\Omega\left( \frac{N^3 q(1 - q)}{\varepsilon_0^2 \rho^2} \right) = \Omega\left( \frac{N^3(1 - \gamma)}{[(1 - \gamma)^2 \varepsilon]^2 \rho^2} \right) = \Omega\left( \frac{N^3}{(1 - \gamma)^3 \varepsilon^2 \rho^2} \right). \qquad \square$$

## D.5 Replicable Policy Estimation Lower Bound

In order to prove a lower bound on locally random replicable policy estimation, we first describe a locally random replicable algorithm that estimates the state-action function $Q^\pi$ for a given (possibly randomized) policy $\pi$. Recall $N := \sum_{s \in \mathcal{S}} |\mathcal{A}^s|$ denotes the number of state-action pairs.

**Lemma D.17.** *Let $\varepsilon, \rho \in (0,1)$ and $\delta \in (0, \rho/3)$. Suppose $\pi$ is an explicitly given randomized policy. There is a locally random $\rho$-replicable algorithm that outputs an estimate $\widehat{Q}^\pi$ of $Q^\pi$ such that with probability at least $1 - \delta$, $\|\widehat{Q}^\pi - Q^\pi\|_\infty \le \varepsilon$. Moreover, the algorithm has sample complexity*

$$\tilde{O}\left(\left(\frac{|\mathcal{S}|N^2}{(1-\gamma)^3 \varepsilon^2 \rho^2} + \frac{N^3}{(1-\gamma)^2 \varepsilon^2 \rho^2}\right) \log \frac{1}{\delta}\right) = \tilde{o}\left(\frac{N^3}{(1-\gamma)^3 \varepsilon^2 \rho^2} \log \frac{1}{\delta}\right).$$

*Proof (Lemma D.17).* Let $\pi : \mathcal{S} \to \Delta(\mathcal{A})$ be explicitly given. First, we construct a single-action MDP $M' = (\mathcal{S}, s_0, \{0\}, P', r', \gamma)$ whose optimal (and only) state-action function $Q'$ is precisely the value function $V^\pi$ of $\pi$.

Let the state space $\mathcal{S}$ and initial state $s_0$ remain the same but replace the action space with a singleton "0". We identify each new state-action pair with the state since only one action exists. Next, define the transition probability

$$P'(s_1 \mid s_0) := \sum_{a \in \mathcal{A}^{s_0}} \pi(s_0, a) \cdot P(s_1 \mid s_0, a).$$

We can simulate a sample from $P'(\cdot \mid s_0)$ by sampling (for free) from the policy, say $a \sim \pi(s_0)$, and then sampling from the generative model $s_1 \sim P(\cdot \mid s_0, a)$. Note this costs a single sample from the generative model. In addition, define the reward function as the expected reward at state $s_0$

$$r'(s_0) := \sum_{a \in \mathcal{A}^{s_0}} \pi(s_0, a) \cdot r(s_0, a).$$

This value can be directly computed given $\pi, r$. Finally, the discount factor $\gamma$ remains the same.

We remark that $Q^*_{M'}(s) = V^\pi_M(s)$ for all $s \in \mathcal{S}$ by construction. Thus the deterministic (non-replicable) $Q$-function estimation algorithm from Theorem D.2 is in fact a deterministic (non-replicable) algorithm that yields an estimate $\widehat{V}^\pi$ of $V^\pi$ such that with probability at least $1 - \delta$,

$$\left\|\widehat{V}^\pi - V^\pi\right\|_\infty \le \frac{\varepsilon'}{2N}$$

$$\varepsilon' := \frac{\varepsilon(\rho - 2\delta)}{\rho + 1 - 2\delta}.$$

Moreover, the sample complexity is

$$\tilde{O}\left(\frac{|\mathcal{S}|N^2}{(1-\gamma)^3 \varepsilon^2 \rho^2} \log \frac{1}{\delta}\right)$$

since the state-action space of $M'$ has size $|\mathcal{S}|$. Without loss of accuracy, assume we clip the estimates to lie in the feasible range $[0, 1/(1-\gamma)]$.

In order to replicably estimate $Q^\pi$, we use the following relation

$$Q^\pi(s, a) = r(s, a) + \gamma \mathbb{E}_{s' \sim P(\cdot|s,a)}\left[V^\pi(s')\right].$$

Note that we only have access to an estimate $\widehat{V}^\pi(s')$ and not $V^\pi(s')$, thus we instead estimate

$$\bar{Q}^\pi(s, a) := r(s, a) + \gamma \mathbb{E}_{s' \sim P(\cdot|s,a)}\left[\widehat{V}^\pi(s')\right].$$

But by Hölder's inequality, we are still guaranteed that with probability at least $1 - \delta$,

$$|\bar{Q}^\pi(s, a) - Q^\pi(s, a)| \le \gamma \|P(\cdot \mid s, a)\|_1 \cdot \left\|\widehat{V}^\pi - V^\pi\right\|_\infty$$

$$= \gamma \left\|\widehat{V}^\pi - V^\pi\right\|_\infty$$

$$\le \frac{\varepsilon'}{2N}$$

for all $s \in \mathcal{S}, a \in \mathcal{A}^s$.

Each expectation of the form $\mathbb{E}_{s' \sim P(\cdot|s,a)}\left[\widehat{V}^\pi(s')\right]$ is simply a bounded statistical query. By an Hoeffding bound, drawing $m$ samples from the generative model $s'_j \sim P(\cdot \mid s,a)$ implies

$$\mathbb{P}\left\{\left|\frac{1}{m}\sum_{j=1}^m \widehat{V}^\pi(s'_j) - \mathbb{E}_{s'}\left[\widehat{V}^\pi(s')\right]\right| > \frac{\varepsilon'}{2N}\right\} \le 2\exp\left(-\frac{2m(1-\gamma)^2\varepsilon'^2}{4N^2}\right).$$

Thus with

$$m = \tilde{O}\left(\frac{N^2}{(1-\gamma)^2\varepsilon'^2}\ln\frac{1}{\delta}\right)$$

trials, the empirical average estimates a single query $\mathbb{E}_{s' \sim P(\cdot|s,a)}\left[\widehat{V}^\pi(s')\right]$ to an accuracy of $\varepsilon'/2N$ with probability at least $1-\delta$. The total number of samples is thus $Nm$.

Combining the $V^\pi$ estimation and Hoeffding bound yields an estimate $\bar{Q}^\pi$ such that

$$\left\|\bar{Q}^\pi - Q^\pi\right\| \le \frac{\varepsilon'}{N}$$

with probability at least $1-2\delta$. Thus we can use the locally random $\rho$-replicable rounding scheme from Theorem C.2 to compute an estimate $\widehat{Q}^\pi$ such that with probability at least $1-2\delta$,

$$\left\|\widehat{Q}^\pi - Q^\pi\right\|_\infty \le \varepsilon.$$

All in all, we described an algorithm that replicably estimates $Q^\pi$ given a policy. The algorithm is locally random since the only internal randomness used to estimate $Q^\pi(s,a)$ occurs at the last locally replicable rounding step. Finally, the total sample complexity of this algorithm is

$$\tilde{O}\left(\left(\frac{|\mathcal{S}|N^2}{(1-\gamma)^3\varepsilon^2\rho^2} + \frac{N^3}{(1-\gamma)^2\varepsilon^2\rho^2}\right)\log\frac{1}{\delta}\right)$$

as desired. □

With Lemma D.17 in hand, we are now ready to prove Theorem D.11, which reduces replicable policy estimation to replicable $Q$-function estimation. We restate the result below for convenience.

**Theorem D.11.** *Let $\varepsilon, \rho \in (0,1)^2$ and $\delta = 1/4$. Suppose $\mathcal{A}$ is a deterministic $\rho$-replicable algorithm that outputs a randomized policy $\pi$ such that $|V^\pi(s) - V^\star(s)| \le \varepsilon$ with probability at least $1 - \delta/12$ for each $s \in \mathcal{S}$. Then $\mathcal{A}$ has a sample complexity of at least*

$$\Omega\left(\frac{N^3}{(1-\gamma)^3\varepsilon^2\rho^2}\right).$$

*Proof (Theorem D.11).* Run $\mathcal{A}$ to output a $\rho$-replicable policy $\pi$ such that

$$|V^\pi(s) - V^\star(s)| \le \varepsilon$$

with probability at least $1 - \delta/12$ for each $s \in \mathcal{S}$.

Applying the algorithm from Lemma D.17 with replicability parameter $\rho$, failure rate $\delta/12$, and $\pi$ as input yields some estimate $\widehat{Q}^\pi$ such that

$$\left|\widehat{Q}^\pi(s,a) - Q^\pi(s,a)\right| \le \varepsilon$$

with probability at least $1 - \delta/12$ for each $s \in \mathcal{S}, a \in \mathcal{A}^s$.

Using the relationship

$$Q^\pi(s,a) = r(s,a) + \gamma\mathbb{E}_{s' \sim P(\cdot|s,a)}\left[V^\pi(s)\right],$$

the triangle inequality, as well as a union bound, we realize that

$$\left|\widehat{Q}^\pi(s,a) - Q^\star(s,a)\right| \le 2\varepsilon$$

with probability at least $1 - \delta/6$ for each $s \in \mathcal{S}, a \in \mathcal{A}^s$. Moreover, $\pi$ is identical across two executions with probability at least $1 - \rho$ by assumption and thus $\widehat{Q}^\pi$ will be identical across two executions with probability at least $1 - 2\rho$.

Remark that the scheme we derived above is a locally random $2\rho$-replicable algorithm for $Q$-estimation. It is locally random since the only source of internal randomness comes from the locally random subroutine in Lemma D.17. Thus the algorithm satisfies the conditions of Theorem D.9 and the lower bound from that theorem applies. In particular, if $m$ denotes the sample complexity of $\mathscr{A}$, then

$$m + \tilde{o}\left(\frac{N^3}{(1-\gamma)^3\varepsilon^2\rho^2}\right) \geq \Omega\left(\frac{N^3}{(1-\gamma)^3\varepsilon^2\rho^2}\right).$$

The LHS is the sample complexity of the scheme we derived courtesy of Lemma D.17 and the RHS is the lower bound from Theorem D.9. It follows that

$$m \geq \Omega\left(\frac{N^3}{(1-\gamma)^3\varepsilon^2\rho^2}\right)$$

as desired. $\qquad\qquad\square$

# E    Omitted Details from Section 4

We now restate and prove Theorem 4.2, a $\rho$-TV indistinguishable SQ oracle for multiple queries.

**Theorem 4.2** (TV Indistinguishable SQ Oracle for Multiple Queries). *Let $\varepsilon, \rho \in (0,1)^2$ and $\delta \in (0, \rho/5)$. Let $\phi_1, \ldots, \phi_d$ be $d$ statistical queries with co-domain $[0,1]$. Assume that we can simultaneously estimate the true values of all $\phi_i$'s with accuracy $\varepsilon$ and confidence $\delta$ using $n(\varepsilon, \delta)$ total samples. Then, there exists a $\rho$-TV indistinguishable algorithm (Algorithm A.1) that requires at most $n\left(\varepsilon\rho/[2\sqrt{8d \cdot \log(4d/\delta)}], \delta/2\right)$ many samples to output estimates $\widehat{v}_1, \ldots, \widehat{v}_d$ of the true values $v_1, \ldots, v_d$ to guarantee that $\max_{i \in [d]}|\widehat{v}_i - v_i| \leq \varepsilon$, with probability at least $1 - \delta$.*

*Proof (Theorem 4.2).* We first argue about the correctness of our algorithm. By assumption, with probability at least $1 - \delta/2$, the oracles provide estimates $\widehat{\mu}_i$'s such that

$$\max_{i \in [d]}|\widehat{\mu}_i - v_i| \leq \frac{\varepsilon\rho}{2\sqrt{8d \cdot \log(4d/\delta)}} \leq \frac{\varepsilon}{2}.$$

We call this event $\mathcal{E}_1$. Fix some $i \in [d]$ and consider the second step where we add Gaussian noise $\mathcal{N}(0, \varepsilon^2)$ to the estimate $\widehat{\mu}_i$ to obtain noisy estimates $\widehat{v}_i$'s. From standard concentration bounds, we know that for $X \sim \mathcal{N}(\mu, \sigma^2), r > 0$

$$\mathbb{P}\{|X - \mu| > r\sigma\} \leq 2e^{-r^2/2}.$$

Plugging in the values

$$\sigma^2 = \frac{\varepsilon^2}{8\log(4d/\delta))}, \qquad\qquad r = \sqrt{2\log\left(\frac{4d}{\delta}\right)},$$

we have that

$$\mathbb{P}\{|\widehat{v}_i - \widehat{\mu}_i| > \varepsilon/2\} \leq \frac{\delta}{2d}.$$

Thus, taking a union bound over $i \in [d]$ we have that

$$\mathbb{P}\left\{\max_{i \in [d]}|\widehat{v}_i - \widehat{\mu}_i| > \varepsilon/2\right\} > \frac{\delta}{2}.$$

We call this event $\mathcal{E}_2$. By taking a union bound over $\mathcal{E}_1, \mathcal{E}_2$ and applying the triangle inequality, we see that

$$\mathbb{P}\left\{\max_{i \in [d]}|\widehat{v}_i - v_i| > \varepsilon\right\} \leq \delta.$$

We now move on to prove the $\rho$-TV indistinguishability guarantees. Consider two executions of the algorithm and let $\widehat{\mu}^{(1)}, \widehat{\mu}^{(2)} \in \mathbb{R}^d$ be the estimates after the first step of the algorithm. We know that, with probability at least $1 - \delta$ over the calls to the SQ oracle across the two executions it holds that

$$\|\widehat{\mu}_1 - \widehat{\mu}_2\|_\infty \leq \frac{\varepsilon\rho}{\sqrt{8d \cdot \log(4d/\delta)}} \,.$$

We call this event $\mathcal{E}_3$ and we condition on it for the rest of the proof. Standard computations [Gupta, 2020] reveal that the KL-divergence between two $d$-dimensional Gaussians $p = \mathcal{N}(\mu_p, \Sigma_p), q = \mathcal{N}(\mu_q, \Sigma_q)$ can be written as

$$D_{\mathrm{KL}}(p\|q) = \frac{1}{2}\left(\log\frac{|\Sigma_q|}{|\Sigma_p|} - d + (\mu_p - \mu_q)\Sigma_q^{-1}(\mu_p - \mu_q) + \mathrm{trace}\left(\Sigma_q^{-1}\Sigma_p\right)\right)$$

In our setting we have

$$p = \mathcal{N}\left(\widehat{\mu}^{(1)}, \frac{\varepsilon^2}{8\log(4d/\delta)} \cdot I_d\right), \qquad q = \mathcal{N}\left(\widehat{\mu}^{(2)}, \frac{\varepsilon^2}{8\log(4d/\delta)} \cdot I_d\right).$$

Plugging these in we have that

$$D_{\mathrm{KL}}(p\|q) = \frac{8\log(4d/\delta)}{\varepsilon^2}\left\|\widehat{\mu}^{(1)} - \widehat{\mu}^{(2)}\right\|_2^2 \leq \rho^2 \,.$$

Thus, using Pinsker's inequality we see that

$$d_{\mathrm{TV}}(p, q) \leq \frac{\rho}{\sqrt{2}}$$

Hence, we can bound the expected TV-distance as

$$\mathbb{E}[d_{\mathrm{TV}}(p, q)] \leq \frac{\rho}{\sqrt{2}} + \delta < \rho \,.$$

$\square$

Next, we prove Theorem 4.8, an implementation of the $\rho$-TV indistinguishable SQ oracle for multiple queries whose internal randomness is designed in a way that also ensures $\rho$-replicability. The result is restated below for convenience.

**Theorem 4.8** (Replicable SQ Oracle for Multiple Queries)**.** *Let $\varepsilon, \rho \in (0, 1)^2$ and $\delta \in (0, \rho/5)$. Let $\phi_1, \ldots, \phi_d$ be $d$ statistical queries with co-domain $[0, 1]$. Assume that we can simultaneously estimate the true values of all $\phi_i$'s with accuracy $\varepsilon$ and confidence $\delta$ using $n(\varepsilon, \delta)$ total samples. Then, there exists a $\rho$-replicable algorithm that requires at most $n(\varepsilon\rho/[4\sqrt{8d\cdot\log(4d/\delta)}], \delta/2)$ many samples to output estimates $\widehat{v}_1, \ldots, \widehat{v}_d$ of the true values $v_1, \ldots, v_d$ with the guarantee that $\max_{i\in[d]}|\widehat{v}_i - v_i| \leq \varepsilon$, with probability at least $1 - \delta$.*

*Proof (Theorem 4.8).* Consider a call to Algorithm A.1 with TV indistinguishability parameter $\rho/2$. Let us call this algorithm $\mathscr{A}$. Notice that given a particular sample, $\mathscr{A}$ is Gaussian and therefore absolutely continuous with respect to the Lebesgue measure. Let $\mathcal{P}$ be the Lebesgue measure over $\mathbb{R}^d$. Let $\mathcal{R}$ be a Poisson point process with intensity $\mathcal{P} \times \mathrm{Leb} \times \mathrm{Leb}$, where $\mathrm{Leb}$ is the Lebesgue measure over $\mathbb{R}_+$ (cf. Theorem C.4).

The learning rule $\mathscr{A}'$ is defined as in Algorithm A.2. For every input $S$ of $\mathscr{A}$, let $f_S$ be the conditional density of $\mathscr{A}$ given the input $S$, let $r := \{(x_i, y_i, t_i)\}_{i\in\mathbb{N}}$ be an infinite sequence of the Poisson point process $\mathcal{R}$, and let $i^\star := \arg\min_{i\in\mathbb{N}}\{t_i : f_S(x_i) > y_i\}$. In words, we consider the pdf of the output conditioned on the input, an infinite sequence drawn from the described Poisson point process, and we focus on the points of this sequence whose $y$-coordinate falls below the curve. The output of $\mathscr{A}'$ is $x_{i^\star}$ and we denote it by $\mathscr{A}'(S; r)$.

We will shortly explain why this is well-defined, except for a measure zero event. The fact that $\mathscr{A}'$ is equivalent to $\mathscr{A}$ follows from the coupling guarantees of this process (cf. Theorem C.4), applied to the single random vector $\mathscr{A}(S)$. We can now observe that, except for a measure zero event, (i) since $\mathscr{A}$ is absolutely continuous with respect to $\mathcal{P}$, there exists such a density $f_S$, (ii) the set over which we are taking the minimum is not empty, (iii) the minimum is attained at a unique point. This means

that $\mathscr{A}'$ is well-defined, except on an event of measure zero[8], and by the correctness of the rejection sampling process [Angel and Spinka, 2019], $\mathscr{A}'(S)$ has the desired probability distribution.

We now prove that $\mathscr{A}'$ is replicable. Since $\mathscr{A}$ is $\rho/2$-TV indistinguishable, it follows that

$$\mathbb{E}_{S,S'}[d_{\mathrm{TV}}(\mathscr{A}(S), \mathscr{A}(S'))] \leq \rho/2.$$

We have shown that $\mathscr{A}'$ is equivalent to $\mathscr{A}$, so we can see that $\mathbb{E}_{S,S'}[d_{\mathrm{TV}}(\mathscr{A}'(S), \mathscr{A}'(S'))] \leq \rho/2$. Thus, using the guarantees of Theorem C.4, we have that for any datasets $S, S'$

$$\mathbb{P}_{r\sim\mathcal{R}}\{\mathscr{A}'(S;r) \neq \mathscr{A}'(S';r)\} \leq \frac{2d_{\mathrm{TV}}(\mathscr{A}'(S), \mathscr{A}'(S'))}{1 + d_{\mathrm{TV}}(\mathscr{A}'(S), \mathscr{A}'(S'))}.$$

By taking the expectation over $S, S'$, we get that

$$\begin{aligned}
\mathbb{E}_{S,S'}\left[\mathbb{P}_{r\sim\mathcal{R}}\{\mathscr{A}'(S,r) \neq \mathscr{A}'(S',r)\}\right] &\leq \mathbb{E}_{S,S'}\left[\frac{2d_{\mathrm{TV}}(\mathscr{A}'(S), \mathscr{A}'(S'))}{1 + d_{\mathrm{TV}}(\mathscr{A}'(S), \mathscr{A}'(S'))}\right] \\
&\leq \frac{2\mathbb{E}_{S,S'}[d_{\mathrm{TV}}(\mathscr{A}'(S), \mathscr{A}'(S'))]}{1 + \mathbb{E}_{S,S'}[d_{\mathrm{TV}}(\mathscr{A}'(S), \mathscr{A}'(S'))]} \\
&\leq \frac{\rho}{1 + \rho/2} \\
&\leq \rho,
\end{aligned}$$

where the first inequality follows from Theorem C.4 and taking the expectation over $S, S'$, the second inequality follows from Jensen's inequality, and the third inequality follows from the fact that $f(x) = 2x/(1 + x)$ is increasing. Now notice that since the source of randomness $\mathcal{R}$ is independent of $S, S'$, we have that

$$\mathbb{E}_{S,S'}\left[\mathbb{P}_{r\sim\mathcal{R}}\{\mathscr{A}'(S;r) \neq \mathscr{A}'(S';r)\}\right] = \mathbb{P}_{S,S',r\sim\mathcal{R}}\{\mathscr{A}'(S;r) \neq \mathscr{A}'(S';r)\}.$$

Thus, we have shown that

$$\mathbb{P}_{S,S',r\sim\mathcal{R}}\{\mathscr{A}'(S;r) \neq \mathscr{A}'(S';r)\} \leq \rho,$$

and so the algorithm $\mathscr{A}'$ is $\rho$-replicable, concluding the proof. $\qquad\square$

*Remark* E.1. The coupling we described in the previous proof is not computable, since it requires an infinite sequence of samples. However, we can execute it approximately in the following way. We can first truncate the tails of the distribution and consider a large enough $d$-dimensional box that encloses the pdf of the truncated distribution. Then, by sampling $\widetilde{O}(\exp(d))$ many points we can guarantee that, with high probability, there will be at least one that falls below the pdf. Even though this approximate coupling is computable, it still requires exponential time.

*Remark* E.2 (Coordinate-Wise Coupling). Since our algorithms add independent Gaussian noise to each of the estimates, a first approach to achieve the coupling using only shared randomness would be to construct a pairwise coupling between each estimate. In the context of multiple statistical query estimation, this would mean that we couple the estimate of the $i$-th query in the first execution with the estimate of the $i$-th query in the second execution. Unfortunately, even though this coupling is computationally efficient to implement it does not give us the desired sample complexity guarantees. To see that, notice that when the TV-distance of estimates across each coordinate is $O(\rho)$, under this pairwise coupling the probability that at least one of the estimates will be different across the two executions is $O(d \cdot \rho)$. However, the TV distance of the $d$-dimensional Gaussians is $O(\sqrt{d} \cdot \rho)$, and this is the reason why the more complicated coupling we propose achieves better sample complexity guarantees. Our results reaffirms the observation that was made by Kalavasis et al. [2023] that the replicability property and the sample complexity of an algorithm are heavily tied to the implementation of its internal randomness, which can lead to a substantial computational overhead.

---

[8]Under the measure zero event that at least one of these three conditions does not hold, we let $\mathscr{A}'(S;r)$ be some arbitrary $d$-dimensional vector in $[0,1]^d$.

# F  Omitted Details from Section 5

We now fill in the missing technical details from Section 5, which proposes an approximately replicable algorithm for optimal policy estimation.

---

**Algorithm F.1** Approximately Replicable Policy Estimation

---

1: $\delta \leftarrow \min\{\delta, \rho_2/2\}$.
2: Run Sublinear Randomized QVI [Sidford et al., 2018a] with confidence $\delta$, discount factor $\gamma$, and error $\rho_1 \cdot \varepsilon \cdot (1-\gamma)/(8\log(|\mathcal{A}|))$.
3: Let $\widehat{Q}$ be the output of the previous step.
4: $\lambda \leftarrow \frac{\log(|\mathcal{A}|)}{\varepsilon/2 \cdot (1-\gamma)}$.
5: For every $s \in \mathcal{S}$, let $\pi(s,a) = \frac{\exp\{\lambda \widehat{Q}(s,a)\}}{\sum_{a' \in \mathcal{A}} \exp\{\lambda \widehat{Q}(s,a')\}}$.
6: Output $\pi$.

---

## F.1  Different Soft-Max Rules

Instead of using the exponential soft-max rule we described in Section 5, one could use a more sophisticated soft-max rule like the piecewise-linear soft-max rule that was developed in Epasto et al. [2020]. The main advantage of this approach is that it achieves a *worst-case $\varepsilon$-approximation*, i.e., it never picks an action that will lead to a cumulative reward that is $\varepsilon$ worse that the optimal one (cf. Definition 5.3). We also remark that this leads to sparse policies when there is only a small number of near-optimal actions.

## F.2  Deferred Proofs

### F.2.1  Proof of Lemma 5.5

We now restate and prove Lemma 5.5, a useful result which quantifies the performance of the soft-max policy in comparison to the optimal policy.

**Lemma 5.5** (Soft-Max Policy vs Optimal Policy). *Let $\varepsilon_1, \varepsilon_2 \in (0,1)^2$. Let $\widehat{Q} \in \mathbb{R}^{\mathcal{S} \times \mathcal{A}}$ be such that $\|\widehat{Q} - Q^\star\| \leq \varepsilon_1$. Let $\hat{\pi}$ be the* ExpSoftMax *policy with respect to $\widehat{Q}$ using parameter $\lambda = \log|\mathcal{A}|/\varepsilon_2$. Then, $\left\|V^{\hat{\pi}} - V^\star\right\|_\infty \leq (2\varepsilon_1 + \varepsilon_2)/(1-\gamma)$.*

*Proof (Lemma 5.5).* Using the guarantees of Lemma 5.4 we have that

$$\max_{a \in \mathcal{A}} \widehat{Q}(s,a) \leq \sum_{a \in \mathcal{A}} \hat{\pi}(\bar{s},a) \cdot \widehat{Q}(s,a) + \varepsilon_2 \,.$$

We can assume without loss of generality that $\pi^\star$ is a deterministic policy due to the fundamental theorem of RL. Fix an arbitrary $s \in \mathcal{S}$. By the fact that $\max_{s \in \mathcal{S}, a \in \mathcal{A}} |\widehat{Q}(s,a) - Q^\star(s,a)| \leq \varepsilon_1$, we see that

$$
\begin{aligned}
Q^\star(s, \pi(s)) &\leq \max_{a \in \mathcal{A}} \widehat{Q}(s,a) + \varepsilon_1 \\
&\leq \sum_{a \in \mathcal{A}} \hat{\pi}(\bar{s},a) \cdot \widehat{Q}(s,a) + \varepsilon_1 + \varepsilon_2 \\
&\leq \sum_{a \in \mathcal{A}} \hat{\pi}(\bar{s},a) \cdot Q^\star(s,a) + 2\varepsilon_1 + \varepsilon_2
\end{aligned}
$$

An equivalent way to write the previous inequality is

$$r(s, \pi^\star(s)) + \gamma \sum_{s' \in \mathcal{S}} V^\star(s') \cdot P_M(s'|s, \pi^\star(s))$$

$$\leq \sum_{a \in \mathcal{A}} \hat{\pi}(s,a) \cdot \left( r(s,a) + \gamma \sum_{s' \in \mathcal{S}} V^\star(s') \cdot P_M(s'|s,a) \right) + 2\varepsilon_1 + \varepsilon_2$$

which implies that

$$r(s, \pi^\star(s)) - \sum_{a \in \mathcal{A}} \widehat{\pi}(s, a) \cdot r(s, a)$$

$$\leq \gamma \left( \sum_{a \in \mathcal{A}} \widehat{\pi}(s, a) \cdot \left( \sum_{s' \in \mathcal{S}} V^\star(s') \cdot P_M(s'|s, a) \right) - \sum_{s' \in \mathcal{S}} V^\star(s') \cdot P_M(s'|s, \pi^\star(s)) \right)$$

$$+ 2\varepsilon_1 + \varepsilon_2 \,. \tag{1}$$

Let $\bar{s} \in \arg\max_{s \in \mathcal{S}} V^\star(s) - V^{\widehat{\pi}}(s)$. Then, we have that

$$V^\star(\bar{s}) - V^{\widehat{\pi}}(\bar{s})$$

$$\leq r(\bar{s}, \pi^\star(\bar{s})) - \sum_{a \in \mathcal{A}} \widehat{\pi}(\bar{s}, a) \cdot r(\bar{s}, a)$$

$$+ \gamma \left( \sum_{s' \in \mathcal{S}} V^\star(s') \cdot P_M(s'|\bar{s}, \pi^\star(\bar{s})) - \sum_{a \in \mathcal{A}} \widehat{\pi}(s, a) \cdot \left( \sum_{s' \in \mathcal{S}} V^{\widehat{\pi}}(s') \cdot P_M(s'|\bar{s}, a) \right) \right) \,.$$

Bounding the first two terms of the RHS using Equation (1) for $s = \bar{s}$ we see that

$$V^\star(\bar{s}) - V^{\widehat{\pi}}(\bar{s})$$

$$\leq \gamma \left( \sum_{a \in \mathcal{A}} \widehat{\pi}(\bar{s}, a) \cdot \left( \sum_{s' \in \mathcal{S}} V^\star(s') \cdot P_M(s'|\bar{s}, a) \right) - \sum_{s' \in \mathcal{S}} V^\star(s') \cdot P_M(s'|\bar{s}, \pi^\star(\bar{s})) \right)$$

$$+ \gamma \left( \sum_{s' \in \mathcal{S}} V^\star(s') \cdot P_M(s'|\bar{s}, \pi^\star(\bar{s})) - \sum_{a \in \mathcal{A}} \widehat{\pi}(\bar{s}, a) \cdot \left( \sum_{s' \in \mathcal{S}} V^{\widehat{\pi}}(s') \cdot P_M(s'|\bar{s}, a) \right) \right)$$

$$+ 2\varepsilon_1 + \varepsilon_2 \,.$$

After we cancel out some terms on the RHS we conclude that

$$V^\star(\bar{s}) - V^{\widehat{\pi}}(\bar{s})$$

$$\leq \gamma \left( \sum_{a \in \mathcal{A}} \widehat{\pi}(\bar{s}, a) \cdot \left( \sum_{s' \in \mathcal{S}} \left( V^\star(s') - V^{\widehat{\pi}}(s') \right) \cdot P_M(s'|\bar{s}, a) \right) \right) + 2\varepsilon_1 + \varepsilon_2$$

$$\leq \gamma \left( \sum_{a \in \mathcal{A}} \widehat{\pi}(\bar{s}, a) \cdot \left( \sum_{s' \in \mathcal{S}} \left( V^\star(\bar{s}) - V^{\widehat{\pi}}(\bar{s}) \right) \cdot P_M(s'|\bar{s}, a) \right) \right) + 2\varepsilon_1 + \varepsilon_2$$

$$= \gamma \left( V^\star(\bar{s}) - V^{\widehat{\pi}}(\bar{s}) \right) + 2\varepsilon_1 + \varepsilon_2 \,.$$

The second inequality follows from the fact that $\bar{s} \in \arg\max_{s \in \mathcal{S}} V^\star(s) - V^{\widehat{\pi}}(s)$ and the equality follows from the fact that $\sum_{a \in \mathcal{A}} \widehat{\pi}(\bar{s}, a) = 1, \sum_{s' \in \mathcal{S}} P_M(s'|\bar{s}, a) = 1, \forall a \in \mathcal{A}$. Rearranging, we see that

$$V^\star(\bar{s}) - V^{\widehat{\pi}}(\bar{s}) \leq \frac{2\varepsilon_1 + \varepsilon_2}{1 - \gamma} \,.$$

But the choice of $s \in \mathcal{S}$ was arbitrary, concluding the proof. $\qquad\square$

### F.2.2 Proof of Theorem 5.6

Last but not least, we prove Theorem 5.6, an upper bound on approximately-replicable policy estimation. We restate the result below for convenience.

**Theorem 5.6.** *Let $\alpha \geq 1, \gamma, \delta, \rho_1, \rho_2 \in (0, 1)^4$, and $\varepsilon \in \left(0, (1 - \gamma)^{-1/2}\right)$. There is a $(\rho_1, \rho_2)$-approximately replicable algorithm $\mathcal{A}$ with respect to the Renyi divergence $D_\alpha$ such that given access to a generator $G$ for any MDP $M$, it outputs a policy $\widehat{\pi}$ for which $\|V^{\widehat{\pi}} - V^\star\|_\infty \leq \varepsilon$ with probability at least $1 - \delta$. Moreover, $\mathcal{A}$ has time and sample complexity $\widetilde{O}\left(N \log(1/\min\{\delta, \rho_2\})/[(1 - \gamma)^5 \varepsilon^2 \rho_1^2]\right)$.*

*Proof (Theorem 5.6).* First, we argue about the correctness of the approach. Using Theorem D.2 with parameters $\gamma, \delta/2, \varepsilon_1 = \rho_1 \cdot \varepsilon \cdot (1-\gamma)/(8\log(|\mathcal{A}|))$, we can estimate some $\widehat{Q}_1$ such that $||\widehat{Q}_1 - Q^\star||_\infty \le \varepsilon_1$, with probability at least $1 - \delta/2$. We call this event $\mathcal{E}_1$ and we condition on it. Since we use the exponential mechanism with parameter $\lambda = \log(|\mathcal{A}|)/(\varepsilon/2 \cdot (1-\gamma))$ to compute a policy $\widehat{\pi}(s)$, for every state $s \in \mathcal{S}$, Lemma 5.5 guarantees that

$$||V^\star - V^{\widehat{\pi}}||_\infty \le \frac{2\rho_1 \cdot \varepsilon \cdot (1-\gamma)/(8\log(|\mathcal{A}|)) + \varepsilon/2 \cdot (1-\gamma)}{1-\gamma} \le \varepsilon\,.$$

This completes the proof of correctness.

We now proceed with showing the replicability guarantees of this process. Consider two executions of the algorithm and let $\widehat{Q}_1, \widehat{Q}_2$ denote the output of the algorithm described in Theorem D.2 in the first and second run, respectively. Notice that, with probability at least $1 - \min\{\delta, \rho_2\}$ it holds that $||\widehat{Q}_1 - Q^\star||_\infty \le \varepsilon_1, ||\widehat{Q}_2 - Q^\star||_\infty \le \varepsilon_1$. We call this event $\mathcal{E}_2$ and condition on it for the rest of the proof. Thus, by the triangle inequality, we have that

$$||\widehat{Q}_1 - \widehat{Q}_2||_\infty \le 2\varepsilon_1\,.$$

Let $\widehat{\pi}_1, \widehat{\pi}_2$ denote the policy the algorithm outputs in the first and second execution, respectively. Let $s \in \mathcal{S}$ be some arbitrary state. Since the exponential soft-max is $2\lambda$-Lipschitz continuous with respect to $(\ell_\infty, D_\alpha)$ we have that

$$\begin{aligned}
D_\alpha(\pi_1(s)||\pi_2(s)) &\le 2\lambda||\widehat{Q}_1 - \widehat{Q}_2||_\infty \\
&= 2 \cdot \frac{\log(|\mathcal{A}|)}{(\varepsilon/2 \cdot (1-\gamma))} \cdot 2 \cdot \frac{\rho_1 \cdot \varepsilon \cdot (1-\gamma)}{(8\log(|\mathcal{A}|))} \\
&= \rho_1\,.
\end{aligned}$$

Since the event $\mathcal{E}_2$ happens with probability $1 - \rho_2$ we see that the algorithm is $(\rho_1, \rho_2)$-approximately replicable with respect to $D_\alpha$, i.e., the Renyi divergence of order $\alpha$. This concludes the approximate replicability part of the proof.

As a last step, we bound the sample complexity of our algorithm. The stated bound follows directly from Theorem D.2 since we call this algorithm with error parameter $\rho_1 \cdot \varepsilon \cdot (1-\gamma)/(8\log(|\mathcal{A}|))$. $\square$

*Remark* F.1 (Replicability Under TV Distance). It is known that the TV distance of two probability distributions is upper bounded by $D_\infty$. Thus, we can see that Theorem 5.6 provides the same guarantees when we want to establish replicability with respect to the TV distance.

*Remark* F.2 (Sample Complexity Dependence on Replicability Parameters). Notice that the dependence of the number of samples in Theorem 5.6 on the two different replicability parameters of Definition 5.2 is different. In particular, the dependence on $\rho_2$ is $\mathrm{polylog}(1/\rho_2)$, whereas the dependence on $\rho_1$ is $\mathrm{poly}(1/\rho_1)$.

## G   Guarantees under Different Replicability Notions

Since we have studied three different replicability notions in this work, we believe it is informative to discuss the advantages and the drawbacks of each one of them. Our discussion is centered across four different axes: the replicability guarantees that each notion provides, the sample complexity required to satisfy each definition, the running time required to run the underlying algorithms, and the ability to test/verify whether the algorithms have the desired replicability properties.

The definition of Impagliazzo et al. [2022] (Definition 2.8) provides the strongest replicability guarantees since it requires that the two outputs are exactly the same across the two executions. It is also computationally efficient to verify it. Even though it is statistically equivalent to the definition of TV indistinguishability, our results along with the results of Bun et al. [2023], Kalavasis et al. [2023] indicate that there might be a computational separation between these two notions. Moreover, the fact that this notion is so tightly related to the way the internal randomness of the algorithm is implemented is a property that is not exhibited by any other notion of stability we are aware of and can be problematic in some applications.

The definition of TV indistinguishability of Kalavasis et al. [2023] (Definition 4.1) provides strong replicability guarantees, in the sense that someone who observes the outputs of the algorithm

under two executions when the inputs are $S, S'$, cannot distinguish which one between $S, S'$ was responsible for generating this output. Moreover, this definition does *not* depend on the way the internal randomness of the algorithm is implemented. On the other hand, testing whether an algorithm has this property is more subtle compared to Definition 2.8. In the case of the Gaussian mechanism-based algorithms we discuss in this work the following holds: if the output of the algorithm is *promised* to be drawn from a Gaussian distribution, it is computationally and statistically efficient to test whether the outputs under two different datasets $S, S'$ are close in TV distance. However, it is not clear how one can test if the outputs are indeed drawn from a Gaussian distribution.

Finally, the notion of approximate replicability (Definition 5.1) we introduce is a further relaxation of the TV indistinguishability property in the following sense: both the replicability definition and TV indistinguishability definition treat the outputs in a "binary" manner, in the sense that they only care whether the outputs are exactly the same across the two executions. This definition takes a more nuanced approach and considers some notion of distance across the outputs that is not binary. As a result, it provides the weakest replicability guarantees, which, however, could be sufficient in most RL applications. Moreover, as our results indicate, there might be some inherent advantage in terms of the sample complexity required to achieve this notion compared to (strict) replicability or TV indistinguishability, which can be crucial in RL applications with large state-action space. Moreover, similarly as with the replicability definition, it is also efficient to test whether an algorithm has this property or not.

To sum up, even though we have not completely characterized the sample complexity and computational complexity of each definition we believe that the following is the complete picture: the replicability property is statistically equivalent to the TV indistinguishability property and the approximate replicability property has sample complexity that is smaller by a factor of $N$. Moreover, we believe that there is a computational gap between the notions of replicability and TV indistinguishability. We underline that under Conjecture D.8, the results of our work give a complete characterization[9] of the sample complexity of these problems with respect to $N$.

---

[9]Up to poly-logarithmic factors.

