# OpenReview forum: "Replicability in Reinforcement Learning"
_NeurIPS.cc/2023/Conference — NeurIPS 2023 poster_

### Official Review · Reviewer_1f6r · 2023-07-01

**Soundness:** 4 excellent
**Presentation:** 3 good
**Contribution:** 3 good
**Rating:** 6
**Confidence:** 4

**Summary:**

This work studies the question of reproducibility in reinforcement learning (RL). They define reproducibility as an algorithm returning the same policy on two different random draws from the environment, with probability at least $\rho$. In the generative model setting, they show that there exists an algorithm which is $\rho$-reproducible, returns an $\epsilon$-optimal policy with probability at least $1-\delta$, and collects at most $O(\frac{N^3 \log 1/\delta}{(1-\gamma)^5 \epsilon^2 \rho^2})$ samples, for $N$ the number of state-actions. They also show a lower bound that this is tight (up to a factor of $1/(1-\gamma)^2$). As the $N^3$ complexity could be prohibitively large in many cases, they show that by relaxing the definition of reproducibility somewhat, they are able to obtain a complexity that scales with $N$ instead.

**Strengths:**

1. To my knowledge, the setting is novel: I do not think the question of reproducibility has been previously considered in the RL literature. Furthermore, the question of algorithmic reproducibility has in general seen attention recently in the broader machine learning literature, so there is general interest in the setting.
2. The results paint a fairly complete picture of the problem, with nearly matching upper and lower bounds. In addition, they show that the $N^3$ dependence can be reduced by considering a slightly relaxed formulation of the problem.

**Weaknesses:**

1. The paper does not clearly motivated why we should care about reproducibility in RL. While reproducibility in science in general is an issue, in RL typically the goal is simply to find a policy that (approximately) maximizes the reward, and we don’t necessarily care whether two runs of the same algorithm produce identical policies or not, as long as they have similar performance. The paper does not give clear justification for why this problem is important, but I think this is necessary given the novelty of it.
2. Definition 2.6 should make clear that the algorithm is able to request which state-actions the $n(\epsilon,\delta)$ are from (if it is indeed able to). As it currently reads, it is ambiguous if the algorithm can request which state-actions they are from, or if it is simply given $n(\epsilon,\delta)$ samples from the generator—the latter option is clearly not possible since if all samples are from the same state-action it is not possible to learn a good policy.
3. Definition 2.9 is also somewhat unclear for a similar reason. My reading is that $\bar{S},\bar{S}’ \sim G$ means that samples from a particular set of state-actions are generated from $G$. Would this exclude the case when all the samples are coming from the state-action? Also, does it exclude an adaptive algorithm which chooses which state-actions to sample from based on the data it has already seen?
4. Using the notation $\bar{r}$ for the randomness of the algorithm is somewhat confusing as $r$ is already used for the reward.
5. I understand that space is limited, but if possible it would be helpful to give some more explanation for the algorithm, in particular the replicable rounding procedure, since this is non-standard in the RL literature. Some intuition on why the $N^3$ dependence arises would also be helpful.

**Questions:**

The primary question I have is on motivating the problem setting, as I stated in the weaknesses section.

**Limitations:**

Yes.

---

> ### Author Rebuttal · Authors · 2023-08-09
>
> We would like to thank the reviewer for their constructive comments.
>
> > The paper does not clearly motivated why we should care about reproducibility in RL.. but I think this is necessary given the novelty of it.
>
> The reviewer raises an important point. In many applications, like mean estimation, correctness of the answer ensures some form of replicability. In the RL setting, we can show that  policies that are derived from two sets of i.i.d. data will yield rewards that are $\varepsilon$-close to one another. However, in many RL applications we set up the rewards as a proxy that will lead to a good policy, i.e. the object of interest is the policy itself and not necessarily its reward. Unfortunately, "standard" RL algorithms do not achieve replicability in the space of policies. This is because most of them are based on the Bellman operator, which is essentially a max operator, and it is very brittle.
>
> As a concrete example, Yu et al. [2019] state that RL applied towards “a medical or clinical treatment regime is composed of a sequence of decision[s] to determine the course of [action] such as treatment type, drug dosage, or re-examination timing … with a goal of promoting the patient’s long-term benefits.” In such cases, where the output we mostly care about is the policy and not the numerical value of the reward, replicability in the policy estimation is crucial. Indeed, practitioners would not trust the results of algorithms that vary significantly when executed on (potentially different) datasets that are obtained using the same sampling process from the same underlying population.
>
> Also, note that replicability within the applied RL community is not a novel concept. Numerous studies such as that of Henderson et al. [2019] and Lynnerup et al. [2019] have attempted to tackle this issue from a methodological perspective. These works provide a series of recommended steps in order to perform replicable RL research. From an algorithmic point of view, the previous works have attempted to minimize the differences in the input of the algorithm since RL algorithms are notoriously sensitive to hyperparameters, the environment, etc. We take a different approach and strive to design algorithms that are inherently more stable, therefore attacking the problem at its root.
>
> > Definition 2.6 should make clear that the algorithm is able to request which state-actions the $n(\epsilon,\delta)$  are from (if it is indeed able to). As it currently reads, it is ambiguous if the algorithm can request which state-actions they are from, or if it is simply given  $n(\epsilon,\delta)$ samples from the generator—the latter option is clearly not possible since if all samples are from the same state-action it is not possible to learn a good policy.
>
> We would like to thank the reviewer for bringing this up. The algorithm is indeed able to specify the state-action pairs. We will clarify that in the next version of our draft.
>
> > Definition 2.9 is also somewhat unclear for a similar reason. My reading is that $S, \bar{S} \sim G$
>  means that samples from a particular set of state-actions are generated from $G$. Would this exclude the case when all the samples are coming from the state-action? Also, does it exclude an adaptive algorithm which chooses which state-actions to sample from based on the data it has already seen?
>
> We would like to thank the reviewer for bringing this up, similarly as before the algorithm is able to select which states-actions it will get samples from. Our algorithms are not adaptive and, intuitively, we do not see how adaptivity could help with replicability. We will clarify that in the next version of our draft.
>
> > Using the notation $\bar{r}$  for the randomness of the algorithm is somewhat confusing as  is already used for the reward.
>
> We thank the reviewer for the constructive comment and will change the notation to use $\xi$ instead.
>
> > I understand that space is limited, but if possible it would be helpful to give some more explanation for the algorithm, in particular the replicable rounding procedure, since this is non-standard in the RL literature. Some intuition on why the $N^3$ dependence arises would also be helpful.
>
> We agree with the reviewer's comment and will include the following explanation in the next version of our manuscript.
> Let $Q^\star$ be the optimal $Q$-function. The replicable rounding procedure is based on a rounding scheme that appeared in [Impagliazzo et al., 2022]. It takes as input two $Q$-vectors $Q_1, Q_2$ with the promise that $||Q_1 - Q_2||_\infty \leq \rho \cdot \epsilon,$ where $\rho, \epsilon$ are the replicability, accuracy parameters respectively. For that, we need $O(N/(\epsilon^2 \rho^2))$ many samples. Then, we consider each coordinate $(s,a)$ separately. We discretize the interval $[0,1/(1-\gamma)]$ that $Q_1(s,a), Q_2(s,a)$ belong to by first consider a random offset $r \sim U[0,\epsilon]$ and then increments of length $\epsilon$. Importantly, this random offset is shared across the two executions. The size of the intervals that we choose for the rounding guarantee that the $|Q_1(s,a) - Q^\star(s,a)| \leq \epsilon, |Q_2(s,a) - Q^\star(s,a)| \leq \epsilon$ and because of the shared random offset we can show that, with probability $1-\rho$, $Q_1(s,a) = Q_2(s,a)$. Because we need to take a union-bound over the $N$ state-action pairs, we need to set the replicability parameter to be $\rho/N$. Since the dependence on $\rho$ is quadratic, we get the extra $N^2$ factor in the sample complexity.
>
> Yu et al. [2019]: Chao Yu, Jiming Liu, and Shamim Nemati: Reinforcement learning in healthcare: a survey.
>
> Henderson et al. [2019]: Peter Henderson, Riashat Islam, Philip Bachman, Joelle Pineau, Doina Precup, David Meger: Deep reinforcement learning that matters.
>
> Lynnerup et al. [2019]: Nicolai A. Lynnerup, Laura Nolling, Rasmus Hasle, John Hallam: A survey on reproducibility by evaluating deep reinforcement learning algorithms on real-world robots.

---

> > ### Comment · Reviewer_1f6r · 2023-08-16
> > **Reply to rebuttal**
> >
> > I would like to thank the authors for their response. I will keep my score as is.

---

### Official Review · Reviewer_GbLw · 2023-07-07

**Soundness:** 3 good
**Presentation:** 3 good
**Contribution:** 3 good
**Rating:** 7
**Confidence:** 4

**Summary:**

The paper studies replicable reinforcement learning algorithms in the tabular MDP setting with an oracle generative model. The paper gives the first lower-bound for the sample complexity of a $\rho$-replicable $(\varepsilon, \delta)$-optimal algorithm. To obtain this lower-bound, the paper first builds an information-theoretical lower-bound for the sample complexity of estimating $N$ independent possibly biased coins, and then reduces the multiple coin estimation problem to some leveled tabular MDP learning with an oracle generative model. By proposing an algorithm (which is basically running an existing RL algorithm for tabular MDPs with a generative model with some carefully designed precision parameters, and then properly rounding the value function) with $\tilde O(N^3)$ dependency of $N$ in the sample complexity upper-bound, the paper shows that the proposed sample complexity lower-bound is indeed sharp in $N$.

The paper also studies a weaker replicability formulation called approximate-replicability and gives an algorithm with $\tilde O(N)$ sample complexity.

**Strengths:**

The paper is clearly written and easy to follow for readers with background on recent theoretical RL works. The given sample complexity lower-bound in $N$ is tight. The proposed algorithms in both formulations are built upon existing oracle algorithms and simple post-processing and hence easy to understand.

**Weaknesses:**

In my opinion, the structure of the paper can be slightly improved. The reductions between the optimal Q-function estimation and the optimal policy, the multiple coin estimation problem and RL on tabular MDPs are at a high level easy to understand. The author may consider compress these contents and move some details of the important technical steps (e.g., the replicable rounding step and how to develop the sample complexity lower-bound for the replicable multiple coin estimation problem) to the main text, in order to bring more insight to the readers after reading the main text.

**Questions:**

Can you provide some insights and comments on the replicability of RL in the pure exploration setting or episodic regret minimization setting (hence both without discounting factors)?

**Limitations:**

The paper considers RL on tabular MDPs with a generative model, it would be more exciting to see results on weaker settings.

---

> ### Author Rebuttal · Authors · 2023-08-09
>
> We would like to thank the reviewer for their constructive comments.
>
> > In my opinion, the structure of the paper can be slightly improved. The reductions between the optimal Q-function estimation and the optimal policy, the multiple coin estimation problem and RL on tabular MDPs are at a high level easy to understand. The author may consider compress these contents and move some details of the important technical steps (e.g., the replicable rounding step and how to develop the sample complexity lower-bound for the replicable multiple coin estimation problem) to the main text, in order to bring more insight to the readers after reading the main text.
>
> We thank the reviewer for the comment. We plan to make these changes in the next version of our draft. Moreover, if the paper gets accepted, we will utilize the extra space to elaborate more on these points.
>
> > Can you provide some insights and comments on the replicability of RL in the pure exploration setting or episodic regret minimization setting (hence both without discounting factors)?
>
> This is indeed an important area for future research. In these settings, the agent will need to keep track of a potential “model” of the world by performing some Upper Confidence Bound (UCB) type of update. The crucial step to ensure replicability is that, by sharing randomness, the learner will be able to update the “model” of the world in the same way across the two executions (e.g., by doing some variant of randomized rounding in the UCB-type of updates). The main difficulty here is to figure out how to come up with a rounding scheme that on the one hand ensures replicability and on the other hand yields low regret/small number of episodes of exploration.
>
> > The paper considers RL on tabular MDPs with a generative model, it would be more exciting to see results on weaker settings.
>
> While we agree with the point that the reviewer raised, our main goal in this work was to provide a comprehensive treatment of the tabular setting, which is the most fundamental setting in RL. We hope and believe that our work will inspire further research in more realistic settings like the linear function approximation setting and the episodic setting. We think that the ideas we have developed in our work, like the replicable $Q$-function estimation (under the different notions of replicability we consider), will be useful in this line of work.

---

> > ### Comment · Reviewer_GbLw · 2023-08-19
> > **Thanks for the rebuttal**
> >
> > The rebuttal looks good to me. I will keep my score.

---

### Official Review · Reviewer_o1wo · 2023-07-11

**Soundness:** 3 good
**Presentation:** 3 good
**Contribution:** 2 fair
**Rating:** 6
**Confidence:** 3

**Summary:**

The paper makes a significant contribution by introducing a theoretical study of replicability in reinforcement learning. It focuses on discounted tabular Markov Decision Processes (MDPs) with generative models and explores two definitions of replicability: exact and approximate versions. For the exact version of replicability, the authors provide a lower bound and propose an algorithm that achieves near-optimal performance, with the exception of dependence on the discount factor $\gamma$. In the case of the approximate version of replicability, the authors show an improved sample complexity.

**Strengths:**

- The paper provides a comprehensive study on the theory of replicability, offering insights into both the exact and approximate versions of replicability. Notably, for the exact version, the authors present both upper and lower bounds, which align closely.

- The concept of replicability in reinforcement learning is highly intriguing, and this paper serves as a pioneering work in advancing the study of theory of replicability in RL. By introducing and delving into this concept, the authors open up new avenues for exploring replicability and its implications in RL research.

- Technically, the paper exhibits strong foundations and analysis, showcasing a solid understanding of the subject matter. The presentation of the research is clear and well-motivated, ensuring that readers can grasp the significance and implications of the findings effectively.

**Weaknesses:**

- While the topic of studying replicability in RL (Reinforcement Learning) is intriguing, I have reservations about its significance. For instance, if there exists a single optimal policy, it logically follows that all RL algorithms should eventually converge to the same policy. Hence, replicability is inherently implied.

- I understand that this is primarily a theoretical paper; however, it would be advantageous to include some experimental analysis. Specifically, exploring the behavior of classical RL algorithms in the tabular case with regards to replicability would provide valuable insights.

**Questions:**

How do classic RL algorithms in the tabular case behave concerning the two notions of replicability discussed in the paper?



**Limitations:**

The authors have addressed the limitations.

---

> ### Author Rebuttal · Authors · 2023-08-09
>
> We would like to thank the reviewer for their constructive comments.
>
> > While the topic of studying replicability in RL (Reinforcement Learning) is intriguing, I have reservations about its significance. For instance, if there exists a single optimal policy, it logically follows that all RL algorithms should eventually converge to the same policy. Hence, replicability is inherently implied.
>
> While we agree with the reviewer that if there is a single optimal policy, replicability is inherently implied, we believe that only a very small set of RL problems fall into this category. On top of that, if we denote by $\delta$ the gap in the reward of the optimal policy and the second best policy, the number of samples we need to find an exact optimal policy scales as $O(1/\delta^2)$. Moreover, if there are two or more optimal policies, then replicability is not trivially implied. In most applications it is the case that either the set of optimal policies is not a singleton, or that the difference in the utility of the optimal policy and the second best policy is so small that we are not willing to pay $O(1/\delta^2)$ in the sample complexity.
>
> Notice that if there are multiple optimal policies, a naive approach is to add random noise to the rewards. This will ensure that, with probability 1, the optimal policy will be unique. Nevertheless, the gap between the best policy and the second best policy will be very small, which will make it prohibitive to pay the required sample complexity in order to be able to detect it.
>
> > I understand that this is primarily a theoretical paper; however, it would be advantageous to include some experimental analysis. Specifically, exploring the behavior of classical RL algorithms in the tabular case with regards to replicability would provide valuable insights.
>
> As the reviewer correctly pointed out, our main focus in this paper is a mathematical treatment of replicability in RL. Nevertheless, we are working on experiments which we will add to the next version of our paper. Let us comment on how classical RL algorithms in the tabular setting behave with regards to the replicability notions we consider. Most of these algorithms like the ones in [Kearns and Singh, 1999], [Azar et al., 2013], are based on the Bellman operator which is inherently non-replicable. This is because the Bellman operator is, essentially, a max operator over some computed quantities, which makes it very sensitive to estimation errors. Thus, these algorithms do not satisfy the replicability definitions we consider in this work. On the other hand, in Section 4 we show that if instead of the max operator we consider a soft-max version of it, we can get results that are ``replicable’’ under our definition of approximate replicability.
>
> > How do classic RL algorithms in the tabular case behave concerning the two notions of replicability discussed in the paper?
>
> Please see our response to the previous comment.

---

> > ### Comment · Reviewer_o1wo · 2023-08-21
> >
> > Thanks for answering my questions!

---

### Official Review · Reviewer_i28M · 2023-07-20

**Soundness:** 4 excellent
**Presentation:** 4 excellent
**Contribution:** 3 good
**Rating:** 2
**Confidence:** 4

**Summary:**

Reproducibility is a big problem in RL. This paper builds on Impagliazzo (2022) on replicability in learning, and develops a replicability framework for RL. It focuses on a discounted tabular MDP setting with a generative model. The replicability problem is given shared internal randomness, how many samples are needed for a learning algorithm to output the exact same policy with probability (1-$\rho$) across two executions. It shows that $\rho$-replicability requires $O(N^2/\rho^2)$ more samples than would otherwise be needed for an \epsilon-optimal Q function, where N is the number of state-action pairs. This matches the lower bound which they also provide. There is also extensions to approximately replicable policy estimation.

**Strengths:**

1. The paper presents a nice framework for replicability in RL which is related to the reproducibility problem.
2. The paper is well-written, and it is easy to understand.
3. The results are interesting though not surprising.

**Weaknesses:**

My main issue with the paper is the underlying premise that the reproducibility crisis in ML and RL is due to the replication difficulty owing to randomness. It is not! As the authors themselves remark, a lot of the reproducibility issues arise due to the need for hyper-parameters, code not being aligned with the algorithms presented in the papers, training details,  environments not being set up in the same way and results reproducible only on trajectories with specific random seeds, etc. As we all well know, if we want to estimate the mean of a distribution from n samples, we will not get the same estimate from two different experiments. That is why confidence intervals should always be reported: That is the standard way, in some sense of ensuring replicability in statistics. And to get say 95\% confidence intervals, it is fairly easy to calculate the number of samples (or sample trajectories) needed. That calculation I would expect would scale as O(N^2) as well. So the question is what more have we learnt from the results in this paper? I think the results in this paper are essentially confidence interval type calculations just dressed up nicely as $\rho$-replicability.

**Questions:**

Q1. Can the authors through some experimental work justify why their notion of replicability (particularly approximate replicability) is more suitable than reporting say 95% confidence intervals? If not experimental work, could you argue this through a simple example?

Q2. Could you elucidate how the replicability framework you introduce can practically help resolve the reproducibility issues in AI/RL?

**Limitations:**

Please explain how the limitations of your framework can be overcome with further work to make it useful for practically addressing the reproducibility issues in AI/RL?

---

> ### Author Rebuttal · Authors · 2023-08-09
>
> We would like to thank the reviewer for their constructive comments.
>
> > As we all well know... So the question is what more have we learnt from the results in this paper? I think the results in this paper are essentially confidence interval type calculations just dressed up nicely as $\rho$-replicability.
>
> The reviewer raises an interesting and important point. In many applications, like mean estimation, correctness of the answer ensures some form of replicability. In particular, if we consider the mean estimation problem we know that, with high probability, if we estimate the mean on two sets of $O(1/\epsilon^2)$ i.i.d. data then the two answers $x_1, x_2$ will be $\epsilon/2$-close to the true one so, by triangle inequality, they will be $\epsilon$-close to each other. Coming back to the RL setting, a similar calculation shows that the two policies that are derived from two sets of i.i.d. data will yield rewards that are $\varepsilon$-close to one another. However, this does not mean that the two policies will be close under any reasonable notion of ``closeness”. In other words, confidence intervals imply replicability in the value space but not in the policy space. In a lot of RL applications we set up the rewards as a proxy that will lead to a good policy, i.e. the object of interest is the policy itself and not necessarily its reward. In order to achieve replicability in the space of policies, we need to do more work than to just report confidence intervals.
>
> As a concrete example, Yu et al. [2019] state that RL applied towards “a medical or clinical treatment regime is composed of a sequence of decision[s] to determine the course of [action] such as treatment type, drug dosage, or re-examination timing … with a goal of promoting the patient’s long-term benefits.” In such cases, where the output we mostly care about is the policy and not the numerical value of the reward, replicability in the policy estimation is crucial. Indeed, practitioners would not trust the results of algorithms that vary significantly when executed on (potentially different) datasets that are obtained using the same sampling process from the same underlying population.
>
> Also, note that replicability within the applied RL community is not a novel concept. Numerous studies such as that of Henderson et al. [2019] and Lynnerup et al. [2019] have attempted to tackle this issue from a methodological perspective. These works provide a series of recommended steps in order to perform replicable RL research. From an algorithmic point of view, the previous works have attempted to minimize the differences in the input of the algorithm since RL algorithms are notoriously sensitive to hyperparameters, the environment, etc. We take a different approach and strive to design algorithms that are inherently more stable, therefore attacking the problem at its root.
>
> > Can the authors through some experimental work justify why their notion of replicability (particularly approximate replicability) is more suitable than reporting say 95% confidence intervals? If not experimental work, could you argue this through a simple example?
>
> We are working on adding experimental evaluation in the next version of our draft. Let us describe a simple example where our notion of approximate replicability is more suitable than just reporting confidence intervals. Consider the problem of computing a stochastic s-t path in a graph. In this problem, the learner has access to a graph but does not know the cost/weight of each edge. We assume that this cost is associated with some distribution in [0,1]. The agent can get information about the cost by querying the edge. Let us consider a naive solution, where the agent estimates the true mean of every edge with accuracy $\epsilon$ and then returns the shortest path based on the estimated means. We can show that with probability 95% the returned path has a total cost at most $\epsilon$ more than the optimal one. However, it is not hard to see that due to the estimation error, with probability 99.99% each run of the experiment returns a different path. It is reasonable to say that such an experiment is not replicable. Our notion of approximate replicability allows, essentially, the agent to return a distribution over paths, so that i) the two distributions are ``close’’ and ii) if we sample a path from this distribution, then with high probability, its cost will be at most $\epsilon$ more than the optimal one.
>
> >  Could you elucidate how the replicability framework you introduce can practically help resolve the reproducibility issues in AI/RL?
>
> One important practical message our work sends regarding the reproducibility issues in AI/RL is that the max operator leads to results that are brittle and are very sensitive to estimation errors and numerical errors, which make the results non-reproducible. On the other hand, soft-max operators like the one we consider to obtain our approximate replicability results, are much more robust to these types of errors.
>
> > Please explain how the limitations of your framework can be overcome with further work to make it useful for practically addressing the reproducibility issues in AI/RL?
>
> Interesting next steps in this line of work are to consider settings beyond the tabular RL, like the linear function approximation setting. Another direction is to consider the episodic setting and see how we can balance the tradeoff between achieving low regret and computing policies that are ``close’’ across executions.
>
> Yu et al. [2019]: Chao Yu, Jiming Liu, and Shamim Nemati: Reinforcement learning in healthcare: a survey.
>
> Henderson et al. [2019]: Peter Henderson, Riashat Islam, Philip Bachman, Joelle Pineau, Doina Precup, David Meger: Deep reinforcement learning that matters.
>
> Lynnerup et al. [2019]: Nicolai A. Lynnerup, Laura Nolling, Rasmus Hasle, John Hallam: A survey on reproducibility by evaluating deep reinforcement learning algorithms on real-world robots.

---

> > ### Author Response · Authors · 2023-08-20
> > **Reviewer i28M**
> >
> > Dear Reviewer i28M,
> >
> > We would be grateful if you could let us know if our explanation made sense to you. We hope the discussion with other reviewers has also helped address any lingering concerns.
> >
> > Thanks a lot!

---

> > ### Comment · Reviewer_i28M · 2023-08-21
> >
> > I am not convinced replicability in the policy space is a worthy goal, or even possible. We can have two different policies with the same value function. And if we care only about policy replicability, we should do imitation learning, perhaps even just behavior cloning. And the problem of replicability from different datasets due to the distribution shift problem, not randomness.
> >
> > Having considered the results in the manuscript, and the authors' response, I find my original score to be too generous. So I am revising it.

---

> > > ### Author Response · Authors · 2023-08-21
> > >
> > > We respectfully disagree with the reviewer's comments.
> > >
> > > > I am not convinced replicability in the policy space is a worthy goal, or even possible.
> > >
> > > The interest in the recent line of work on the formal study of replicability in ML initiated by Impagliazzo et al., which studies a similar problem as we do, shows that replicability in the space of classifiers/models and not just utility, is a topic of interest in the ML community. The comment that this goal might not even be possible is factually incorrect, since our results show that it is, indeed, possible. In fact, in the case of approximate replicability, we show that this property can be achieved with poly-logarithmic sample complexity overhead. This discussion also contradicts the comment that our results are not surprising, which was raised by the same reviewer.
> > >
> > > > We can have two different policies with the same value function.
> > >
> > > As we explained, in many applications having the same value function is *not* the end goal.
> > >
> > > > And if we care only about policy replicability, we should do imitation learning, perhaps even just behavior cloning.
> > >
> > > We are a bit confused about this comment. Our work provides a black-box transformation to achieve replicability in the policy space, no matter what the underlying algorithm we begin with is. We don't see why restricting ourselves to imitation learning and behavior cloning would provide any benefit. In the case of exact replicability, we provide lower bounds which illustrate that the approach the reviewer is suggesting will not lead to improved sample complexity compared to our current results. We underline again that in the case of approximate replicability, our transformation comes, essentially, at no cost in the sample complexity of the algorithm. We would be happy to provide further clarifications if the reviewer can explain what they mean by their comment.
> > >
> > > > And the problem of replicability from different datasets due to the distribution shift problem, not randomness.
> > >
> > > This is factually incorrect. Even if the datasets are coming from the same distribution, as we have argued in our work and in our rebuttal, the policies that the algorithm outputs can be significantly different. Of course, the problem of replicability becomes even more difficult in the setting of distribution shifts, but this is beyond the scope of our work. We hope and believe that our work will provide the foundation to study the problem of replicability in more complex environments, like the ones that have distribution shifts.

---

> > > > ### Comment · Reviewer_i28M · 2023-08-21
> > > >
> > > > I respectfully strongly disagree. I think the claims above are a bunch of baloney. First, the replicability problem as defined in this paper has nothing to do with the reproducibility in AI/ML crisis. Second, if the optimal action in any state is not unique, there is no way to "replicate" the policy. If that is not the goal (contrary to above claims now), and , then we are really talking about "replicability" in the value function space, in which case I don't really see what the big deal is! Suppose I want to estimate mean of a distribution in two separate iid experiments. If I want the returned results to be within epsilon, it is pretty straightforward to calculate how many samples one would need. At some level, this is what is being done in this paper. And it is neither surprising, not very useful. I am revising my score further.

---

> > > > > ### Author Response · Authors · 2023-08-21
> > > > >
> > > > > The reviewer is being utterly disrespectful. We refuse to entertain conversations that use unprofessional and derogatory terms for our work, like "baloney". The reviewer should stick to the scientific part of the work and avoid such behaviors.
> > > > >
> > > > > > First, the replicability problem as defined in this paper has nothing to do with the reproducibility in AI/ML crisis
> > > > >
> > > > > The reviewer fails to see the point in this line of work that has led to publications in very well-established conferences like STOC/COLT/ICLR/ICML.
> > > > >
> > > > > > Second, if the optimal action in any state is not unique, there is no way to "replicate" the policy.
> > > > >
> > > > > Again, this is factually incorrect and it shows minimal to zero understanding of our paper. We show that it is indeed possible to do that.
> > > > >
> > > > > >  If that is not the goal (contrary to above claims now), and , then we are really talking about "replicability" in the value function space, in which case I don't really see what the big deal is!
> > > > >
> > > > > We are not talking about replicability in the value space, we are talking about replicability in the policy space. We formally prove that this is possible. If the reviewer cannot understand the main results of our paper, it is impossible to have any discussion of scientific merit.
> > > > >
> > > > > > Suppose I want to estimate mean of a distribution in two separate iid experiments. If I want the returned results to be within epsilon, it is pretty straightforward to calculate how many samples one would need.
> > > > >
> > > > > Again, we underline that this is *not* what we are doing. This shows some type of replicability in the value space and not in the policy space. Our work shows that replicability in the policy space is indeed possible, and in the case of the approximate replicability definition, this can be achieved with minimal overhead in the sample complexity.
> > > > >
> > > > > > At some level, this is what is being done in this paper. And it is neither surprising, not very useful. I am revising my score further.
> > > > >
> > > > > The behavior of the reviewer shows that they are acting completely unprofessionally. Adjusting the score based on personal sentiment rather than scientific merit is *not* what the reviewing process is about.

---

> > > > > > ### Comment · Reviewer_i28M · 2023-08-21
> > > > > >
> > > > > > This is a totally unacceptable response.

---

### Decision · Program_Chairs · 2023-09-21

**Decision:**

Accept (poster)

**Comment:**

The paper studies replicability in reinforcement learning. The paper considers discounted tabular Markov Decision Processes (MDPs) with generative models and explores two definitions of replicability: exact and approximate versions. For the exact version of replicability, the authors provide a lower bound and propose an algorithm that achieves near-optimal performance, with the exception of dependence on the discount factor. In the case of the approximate version of replicability, the authors show an improved sample complexity.

While one of the reviewer remains concerned on the definition of the replicability, which can be different from what readers would intuitively imagine, this paper has rigorous definition of the terminology and develops solid theory for author's version replicability, and is thus favored by remaining reviewers. Therefore, we recommend acceptance. It would be appreciated if authors could potentially modify proper contents to avoid the confusion in the discussion.